# Distinct neural contributions to metacognition for detecting, but not discriminating visual stimuli

**Matan Mazor[1]\*, Karl J Friston[1], Stephen M Fleming[1,2,3]**

[1]Wellcome Centre for Human Neuroimaging, University College London, London, United Kingdom; [2]Max Planck UCL Centre for Computational Psychiatry and Aging Research, University College London, London, United Kingdom; [3]Department of Experimental Psychology, University College London, London, United Kingdom

**Abstract** Being confident in whether a stimulus is present or absent (a detection judgment) is qualitatively distinct from being confident in the identity of that stimulus (a discrimination judgment). In particular, in detection, evidence can only be available for the presence, not the absence, of a target object. This asymmetry suggests that higher-order cognitive and neural processes may be required for confidence in detection, and more specifically, in judgments about absence. In a within-subject, pre-registered and performance-matched fMRI design, we observed quadratic confidence effects in frontopolar cortex for detection but not discrimination. Furthermore, in the right temporoparietal junction, confidence effects were enhanced for judgments of target absence compared to judgments of target presence. We interpret these findings as reflecting qualitative differences between a neural basis for metacognitive evaluation of detection and discrimination, potentially in line with counterfactual or higher-order models of confidence formation in detection.

**\*For correspondence:**
mtnmzor@gmail.com

**Competing interests:** The authors declare that no competing interests exist.

## Introduction

When foraging for berries, one first needs to decide whether a certain bush bears fruit or not. Only if berries are detected, can one proceed to examine and classify them into a category - are these raspberries or blackberries? The first is a *detection* task: a decision about whether something is there or not, and the second is a *discrimination* task: a decision about which item is there. For these types of decisions, it is important not only to understand the decision process that leads to deciding present or absent, or raspberries or blackberries, but also our ability to reflect on and estimate the quality of the decision, known as metacognition. For instance, two foragers working together may want to share their confidence in deciding which bush to tackle next (*Bahrami et al., 2010*; *Frith, 2012*).

There is an increasing understanding of the neural basis of confidence in simple decisions, with a network of prefrontal and parietal regions being identified as important for tracking metacognitive beliefs about the accuracy of both perceptual and value-based decisions (see *Domenech and Koechlin, 2015*; *Meyniel et al., 2015*, for reviews). Accordingly, neuropsychological data in humans suggest that damage or impairment of prefrontal function can lead to metacognitive impairments such as noisy or inappropriate confidence judgments (see *Rouault et al., 2018*, for a review). However, in a majority of these cases, the study of confidence has been restricted to discrimination, or deciding whether a stimulus is from category A or B. Despite their ubiquity and importance in decision-making, much less is known about how confidence is formed in detection settings, in which subjects are asked to make a judgment about whether a target stimulus is present or not.

Computational considerations and behavioral findings suggest that computing confidence in detection judgments may differ from computing confidence in the more commonly studied

discrimination tasks. In particular, detection is unique in the landscape of perceptual tasks in that evidence can only be available to support the presence, not the absence, of a target object. This makes confidence ratings in judgments about absence a unique case, where confidence is decoupled from the amount of supporting perceptual evidence. Accordingly, behavioral evidence indicates that metacognitive sensitivity, or the alignment between subjective confidence and objective performance, for judgments about absence is typically impaired compared to metacognitive sensitivity for judgments about presence (*Meuwese et al., 2014*; *Kanai et al., 2010*).

Under one family of models (*first-order models*), confidence in detection judgments is formed in the same way as confidence in discrimination judgments. For example, in evidence-accumulation models, confidence can be evaluated as the distance of the losing accumulator from the threshold at the time of decision (*Vickers, 1979*; *Merkle and Van Zandt, 2006*). Similarly, in models of discrimination confidence based on *Signal Detection Theory* (SDT), decision confidence is assumed to be proportional to the strength of the available evidence supporting the decision, which is modeled as the distance of the perceptual sample from the decision criterion on a strength-of-evidence axis (*Wickens, 2002*, section 5.2). While first-order models are traditionally symmetric, they can be adapted to account for the asymmetry between judgments about presence and absence. For example, *unequal-variance (uv-SDT)* and *multi-dimensional SDT models* account for the inherent difference between presence and absence by making the signal distribution wider than the noise distribution (*Wickens, 2002*, section 3.4), or by assuming a high-dimensional stimulus space, in which the absence of a signal is represented as a distribution centered around the origin (*King and Dehaene, 2014*; *Wickens, 2002*, section 7.2). Importantly, first-order models treat the process of metacognitive evaluation of detection and discrimination as qualitatively similar, with any differences between detection and discrimination emerging from differences in the underlying distributions (uv-SDT), or the mapping between stimulus features and responses (two-dimensional SDT).

In contrast with first-order models of detection confidence, *higher-order models* treat confidence in judgments about target absence as emerging from a distinct, higher-order cognitive process. For instance, in one version of the higher-order approach, confidence in judgments about absence is assumed to be based on counterfactual estimation of the likelihood of a hypothetical stimulus to be detected, if presented. In other words, subjects may be more confident in the absence of a target object when they believe they would not have missed it, based on their global estimation of task difficulty, or on their current level of attention. A similar type of modeling has been successfully employed in studies of memory, to explain how participants form judgments that an item was not presented during the preceding learning phase, based on their counterfactual expectations about remembering an item (*Glanzer and Adams, 1990*). When applied to the comparison of detection and discrimination, this approach predicts that qualitatively distinct cognitive and neural resources will be recruited when judging confidence in detection responses, due to the additional demand on counterfactual and self-monitoring processes, and that this recruitment will be most pronounced for confidence about absence. In particular, the counterfactual account predicts that responses in the frontopolar cortex, a region which has been shown to track counterfactual world states (*Boorman et al., 2009*), will show specificity for confidence judgements when inferring the absence of a target.

To test for such qualitative differences, here we set out to directly compare the neural basis of metacognitive evaluation of detection and discrimination responses within two similar low-level perceptual tasks, while controlling for differences in task performance. In a pre-registered design, we asked whether parametric relationships between subjective confidence ratings and the blood-oxygenation-level-dependent (BOLD) signal in a set of predefined prefrontal and parietal regions of interests (ROIs) would show systematic interaction with task (detection/discrimination) and, within detection, type of response (present/absent). To anticipate our results, we observed a quadratic effect of confidence on regional responses in frontopolar cortex for detection, but not for discrimination judgments. In further whole-brain exploratory analyses, we found stronger confidence-related effects for judgments of absence compared to presence in right temporoparietal junction.

## Results

A total of 35 participants performed two perceptual decision-making tasks while being scanned in a 3T MRI scanner: an orientation discrimination task (*'was the grating tilted clockwise or*

*anticlockwise?'*), and a detection task (*'was any grating presented at all?'*; see **Figure 1**). The discrimination and detection tasks were performed in separate blocks each lasting 40 trials. At the end of each trial, participants rated their confidence in the accuracy of their decision on a 6-point scale. We adjusted the difficulty of the two tasks in a preceding behavioral session to achieve equal performance of around 70% accuracy. At scanning, 10 discrimination and detection blocks were presented in 5 scanner runs.

## Behavioral results

Task performance was similar for detection (75% accuracy, d'=1.48) and discrimination blocks (76% accuracy, d'=1.50). Repeated measures t-tests failed to detect a difference between tasks both in mean accuracy (t(34) = −0.90, p=0.37, $BF_{01}$ = 5.15), and d' ( t(34) = −0.30, p=0.76, $BF_{01}$=7.29), indicating that performance was well matched. Responses were also balanced for the two tasks. The probability of responding YES (target present) in the detection task was 0.49 ± 0.11, and not significantly different from 0.5 (t(34) = −0.39, p=0.70, $BF_{01}$=7.07). The probability of responding CLOCKWISE in the discrimination task was 0.50 ± 0.08, and not significantly different from 0.5 (t(34) = 0.22, p=0.83, $BF_{01}$=7.43).

The distribution of confidence ratings was generally similar between the two tasks and four responses. For all four responses, participants were most likely to report the highest confidence rating compared to any other option. Within detection, a significant difference in mean confidence was observed between YES (target present) and NO (target absent) responses, such that participants were more confident in their YES responses (t(34) = −4.85, p<0.0001; see **Figure 2**). This difference in mean confidence was mostly driven by the higher proportion of maximum confidence ratings in YES responses compared to NO responses (46% of all YES responses compared to 26% of all NO responses, t(34)=5.63, p<0.00001), but persisted even when ignoring the highest ratings (t(34)=2.39, p<0.05).

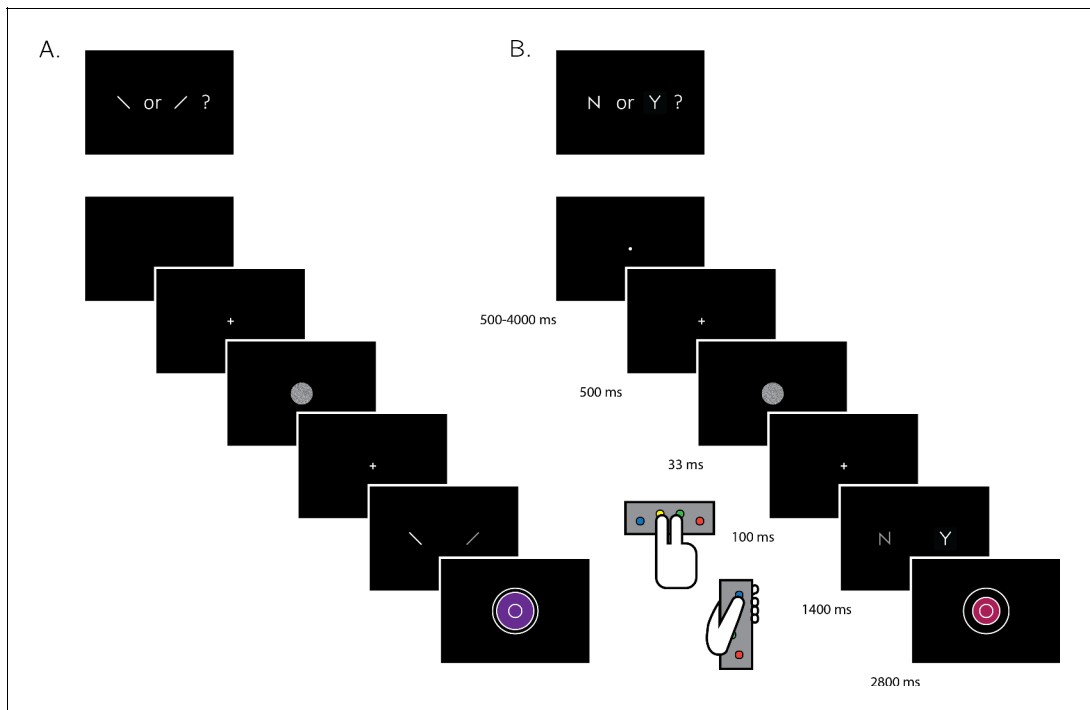

**Figure 1.** Experimental design for discrimination and detection trials. Perceptual decisions were reported using the right index and middle fingers, and confidence ratings were reported using the left thumb. (**A**) In discrimination blocks, participants indicated the orientation of a visual grating (CLOCKWISE or ANTICLOCKWISE). (**B**) In detection blocks, participants indicated whether a grating was embedded in the random noise, or not (YES or NO). Confidence ratings were made by varying the size and color of a circle, with 6 options ranging from small and red to big and blue. For half of the subjects, high confidence was mapped to a small, red circle. For the other half, high confidence was mapped to a big, blue circle. The initial size and color of the circle was determined randomly at the beginning of the confidence rating phase. Participants performed 10 interleaved 40-trial detection and discrimination blocks inside a 3T MRI scanner.

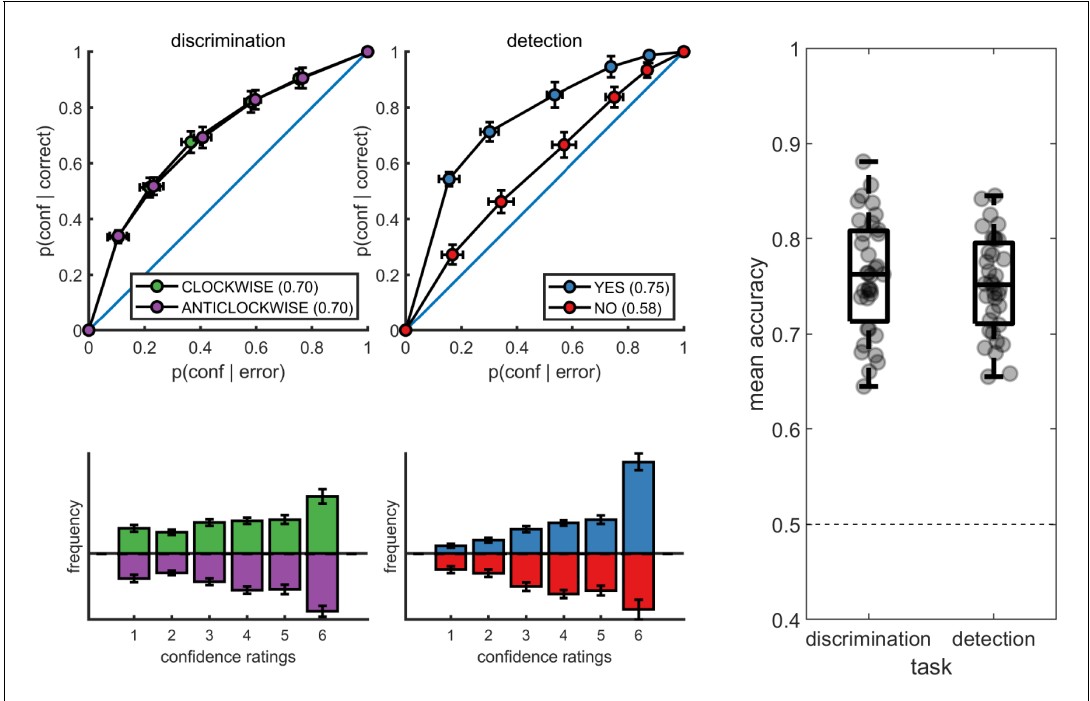

**Figure 2.** Upper panels: response conditional type-2 ROC curves. In parentheses: the mean area under the curve. Lower panels: distribution of confidence ratings for the two tasks and four responses. Right panel: Mean accuracy for both tasks. Error bars represent the standard error of the mean.

Metacognitive sensitivity, quantified as the area under the type-II ROC curve, was significantly higher for YES compared to NO responses (t(34) = 7.83, p<10–8; see *Figure 2*), as expected (*Meuwese et al., 2014*). In other words, confidence ratings about the presence of a target stimulus were more diagnostic of accuracy than ratings about target absence, even though both sets of ratings tended to cover the full range of the scale, from low to high confidence. Taking metacognitive sensitivity following discrimination responses as a baseline, we found that this effect was driven by a decrease in metacognitive sensitivity for NO responses (t(34) = −4.89, p<0.0001), whereas a quantitative increase in metacognitive sensitivity for YES responses compared to discrimination was not significant (t(34)=1.84, p=0.07). No difference was observed in metacognitive sensitivity between the two discrimination responses (CLOCKWISE and ANTICLOCKWISE; t(34) = 0.06, p=0.95, $BF_{01}$=7.6). Taken together, these results are consistent with the previously reported selective asymmetry in the fidelity of metacognitive evaluation following judgments about target absence (*Meuwese et al., 2014*; *Kanai et al., 2010*).

Response times were faster on average for correct responses (849 ± 79 milliseconds) compared to incorrect responses (938 ± 95 milliseconds; t(34)=10.59, p<10^-11 for a paired t-test on the log-transformed response times). Within the detection task, YES responses were significantly faster than NO responses (850 ± 90 milliseconds and 896 ± 103 milliseconds, respectively; t(34)=3.16, p<0.005 for a paired t-test on the log-transformed response times).

## Imaging results
### Parametric effect of confidence
We next turned to our fMRI data to ask whether confidence-related responses were similar or distinct across tasks (detection/discrimination) and response (target present: YES/target absent: NO). We first established the presence of linear confidence-related effects in our a priori ROIs, both across tasks and response types and across correct and incorrect responses, in line with previous findings of 'generic' or task-invariant confidence signals in these regions (*Morales et al., 2018*). Specifically, higher confidence ratings were associated with increased activation in the ventromedial prefrontal cortex (vmPFC), the ventral striatum, and the precuneus. Conversely, activations in the posterior medial frontal cortex (pMFC) were negatively correlated with confidence (see *Figure 3*). For the

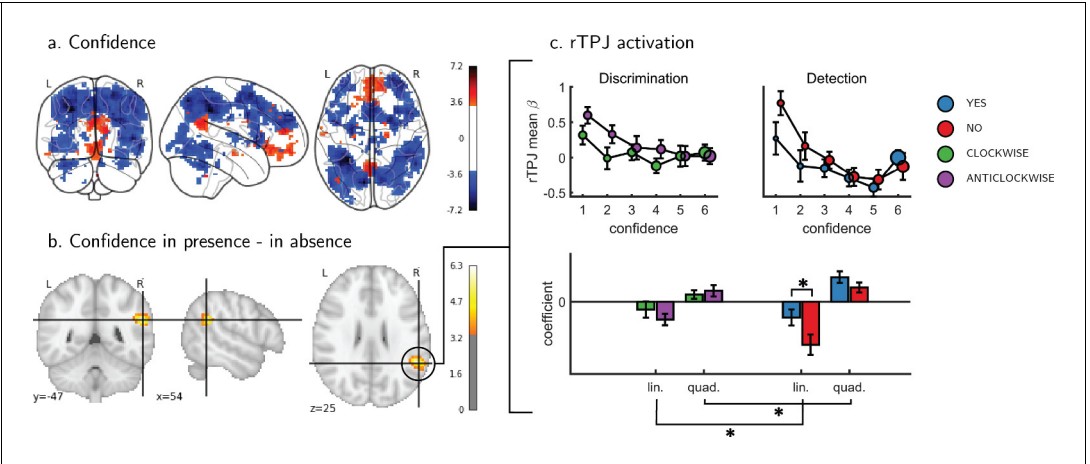

**Figure 3.** Univariate parametric effect of confidence. (a) Glass brain visualization of global effect of confidence, thresholded at the single voxel level for visualization (p<0.001, uncorrected). Negative confidence effect appears in blue, and positive effect in red. (b) Whole brain contrast between confidence in 'target present' (YES) and 'target absent' (NO) detection responses, corrected for family-wise error rate at the cluster level (p<0.05) with a cluster defining threshold of p<0.001, uncorrected. (c) Upper panel: BOLD signal in the rTPJ cluster from panel b as a function of response and confidence. lower panel: mean coefficients of response- and subject-specific multiple linear regression models, predicting rTPJ activation as a linear and quadratic function of confidence. * - p<0.05; uncorrected for multiple comparisons across the four tests. Comparison lines above and below the x axis indicate main effect of response and task, respectively.

confidence effect pattern obtained from the Global-Confidence Design Matrix (GC-DM), see **Appendix 3—figure 1**.

## Interaction of linear confidence effects with task and response

We next asked whether the linear parametric relationship between confidence and BOLD activity differed as a function of task (discrimination vs. detection) and response type (YES vs. NO in detection). In the pMFC, vmPFC, ventral striatum and precuneus ROIs, the parametric effect of confidence failed to show a significant difference between the two tasks (all p-values>0.3), between the two discrimination responses (all p-values>0.24), or between the two detection responses (all p-values>0.09). Similarly, no cluster within the pre-specified frontopolar ROI showed a differential effect of confidence as a function of task or response. We show below that this absence of a linear interaction should not be taken as evidence of absence of differences between detection and discrimination, due to the presence of nonlinear interaction effects. In the next section we first explain the analysis steps we took to uncover nonlinear effects of confidence.

## Interaction of nonlinear confidence effects with task and response

An exploratory whole brain analysis (p<0.05, corrected for multiple comparisons at the cluster-level) revealed no differential confidence effect as a function of task anywhere in the brain. However, within detection, whole-brain analysis revealed that the linear effect of confidence was significantly more negative for NO compared to YES responses in the right temporo-parietal junction (rTPJ: 101 voxels, peak voxel: [54,-46, 26], z = 5.10). To further characterize the nature of the interaction between confidence and response in the rTPJ, we fitted a new design matrix for each task (Categorical-Confidence Design Matrices (post-hoc analysis; CC-DM)) where confidence was represented as a categorical variable with 6 levels instead of one parametric modulator. In contrast to our original design matrix (Main Design Matrix (DM-1)) that assumed a linear effect of confidence, this analysis is agnostic as to the functional form of the confidence effect. We then plotted the mean activation level for each combination of response and confidence level in the rTPJ cluster (see **Figure 3**, panel c).

The categorical-confidence design matrix revealed a positive quadratic effect of confidence on activation levels in the rTPJ, with stronger activation levels for the two extremities of the confidence scale. We confirmed the presence of a significant quadratic effect of confidence in this region by fitting a second-order polynomial to the response-specific confidence curve of each participant (see

Materials and methods). This analysis revealed a main quadratic effect of confidence in this region (t (34) = 5.21, p<0.00001), an effect which was stronger in detection compared to discrimination (t(34) =2.06, p<0.05, d = 0.35). Importantly, the linear interaction of confidence with detection responses remained significant for this quadratic model, establishing that this response-specific effect is not explained by an overall quadratic pattern (t(33)=2.09, p<0.05, d = 0.36 ; see *Figure 3*). More generally, these analyses make clear that linear effects of parametric modulators and their interactions are not exhaustive in their characterization of the confidence-related BOLD response – in this region and potentially in our other ROIs too.

To formally test for such nonlinear differences in the activation profile of other ROIs, we extracted the coefficients from the categorical model for each ROI, and fitted a second-order polynomial to the ensuing confidence-related response. Within our a priori ROIs, no quadratic effect of confidence was observed in the pMFC, the precuneus, the ventral striatum, or the vmPFC (*Appendix 5—figure 1*). In contrast, in all three anatomical subregions of the frontopolar cortex, we found a positive quadratic effect of confidence, with stronger activations for the two extremities of the confidence scale. Strikingly, in both the FPl and the FPm, this positive quadratic effect of confidence was entirely driven by the detection task (FPm: t(34)=3.04, p<0.005, d = 0.51; FPl: t(34)=3.90, p<0.001, d = 0.66; see *Figure 4*). Confidence ratings for the discrimination task however showed a quadratic effect that was not statistically different from zero (FPm: t(34)=-0.54, p=0.59, d = −0.09, $BF_{01}$=6.61; FPl: t(34) =1.42, p=0.16, d = 0.24, $BF_{01}$=2.92). In the FPm, the linear effect of confidence was more negative for detection than for discrimination (t(34) = −2.11, d = −0.36, p<0.05), and within detection, more negative for confidence in judgments about absence (NO responses; t(34) = 2.10, d = −0.36, p<0.05).

Finally, to test for similar quadratic effects of confidence at the whole-brain level, we constructed a new design matrix (in a departure to our pre-registered analysis plan) in which confidence was modeled by a parametric modulator with a polynomial expansion of 2 (Quadratic-Confidence Design Matrix (post-hoc analysis; QC-DM)). Three clusters showed a significantly stronger quadratic effect of confidence in detection compared to discrimination (*Figure 5*). These were located in the right superior temporal sulcus (72 voxels, peak voxel: [60,-43,2], Z = 3.99), pre-SMA (130 voxels, peak voxel: [0,35,47], Z = 4.07), and right frontopolar cortex, overlapping with our FPl and FPm frontopolar anatomical subregions (51 voxels, peak voxel: [9,65,-10], Z = 4.00). Importantly, no region showed stronger quadratic effects of confidence in discrimination compared to detection.

To visualize activity patterns in these regions, we extracted the mean coefficients from the categorical model for these three clusters, and fitted a second-order polynomial separately to each response estimate (see *Figure 5*). In addition to the effect of task on the quadratic effect of confidence in all three clusters, the linear effect of confidence in the right frontopolar cluster was significantly more negative for detection, compared to discrimination (t(34)=-3.13, d = −0.53, p<0.005). For both tasks, inter-subject variability in metacognitive efficiency (measured as meta-d'/d'; *Maniscalco and Lau, 2012*) was not reliably correlated with linear or quadratic parametric effect of confidence in any of the three regions (see Appendix 7).

## Computational models

We next considered alternative computational-level explanations for the detection-specific quadratic activation profile. Specifically, we evaluated how latent model variables or belief states change nonlinearly as a function of confidence in three candidate model architectures (see *Figure 6*): a static 'Signal Detection' model, a 'Dynamic Criterion' model where policy changes as a function of previous perceptual samples, and an 'Attention Monitoring' model in which beliefs about fluctuations in attention inform decisions and confidence judgments. A detailed formal description of the three models is available in the appendix (sections 9, 10 and 11).

First, we consider the static Signal Detection Theory (SDT) model. In SDT models of confidence formation, the log likelihood-ratio between the two competing hypotheses ($LLR = \log \frac{p(x|S_1)}{p(x|S_2)}$) is a useful measure for determining the certainty with which one should commit to a choice. The mapping between the perceptual sample $x$ and the LLR is linear for equal-variance SDT, which is often used to model discrimination, but quadratic for unequal-variance SDT, which is often used to model detection. It then follows that if confidence is proportional to the distance of the sample $x$ from the decision criterion, neuronal populations that represent the relative likelihood of a choice being correct

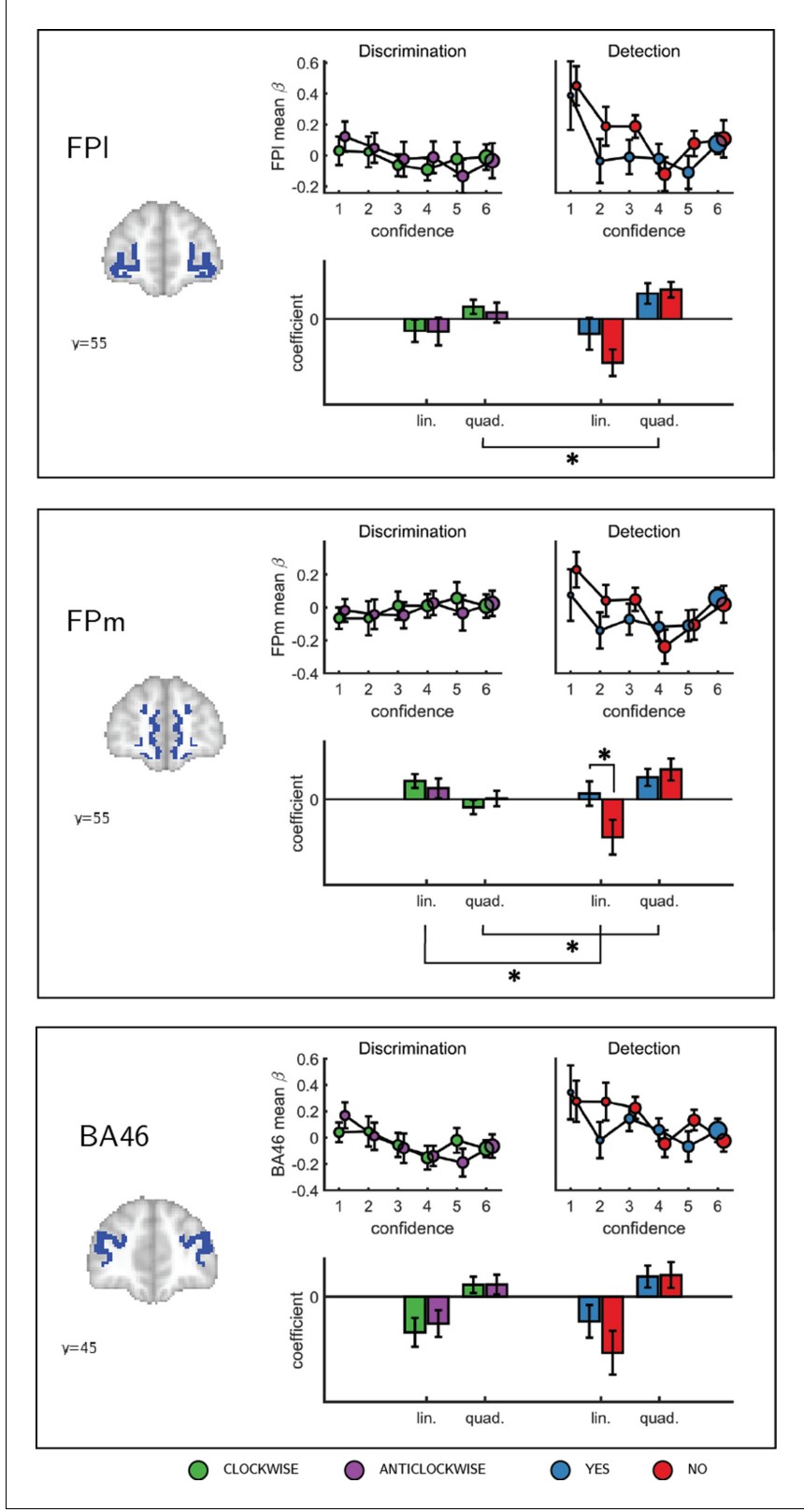

**Figure 4.** Confidence effect as a function of response in the frontopolar cortex separated into its three anatomical subcomponents: FPm, FPl, and BA 46. Same conventions as in *Figure 3c*. * - p<0.05; uncorrected for multiple comparisons. Comparison lines above and below the x axis indicate main effect of response and task, respectively.

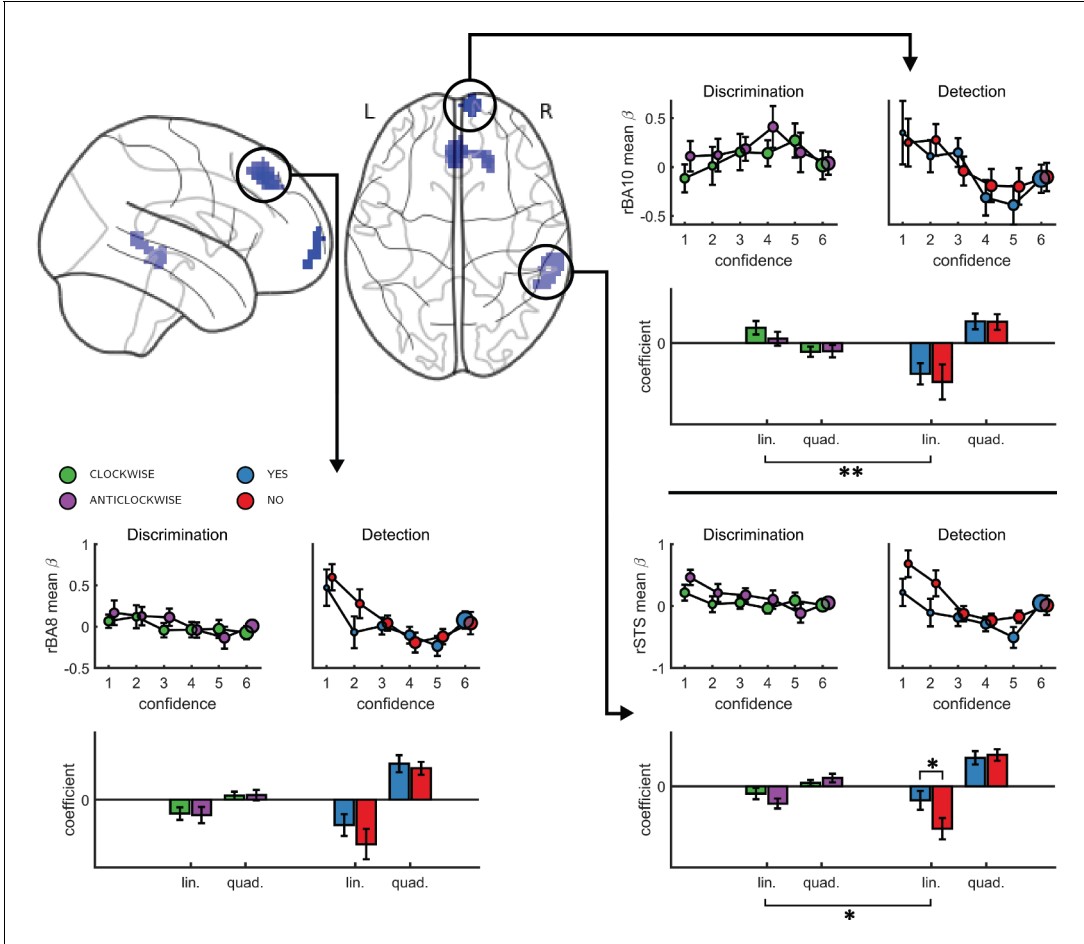

**Figure 5.** Left, top panel: a glass-brain representation of a contrast between the quadratic effects of confidence in detection and in discrimination, whole-brain corrected for family-wise error rate at the cluster-level (p<0.05) with a cluster-defining threshold of p<0.001, uncorrected. Remaining panels: mean betas from the categorical model for each of the four responses and six confidence ratings, for the three indicated clusters. The second-order polynomial coefficients for these estimates are presented below each plot. Significance is only indicated for the linear effects, which are orthogonal to the quadratic contrast used to select the clusters. * - p<0.05; ** - p<0.01. Comparison lines above and below the x axis indicate main effect of response and task, respectively.

(be it LLR or an analogue quantity) will show a quadratic tuning function of confidence in detection and a linear tuning function in discrimination, similar to that observed in FPC, pre-SMA and STS. However, LLR is also expected to scale more strongly with confidence in YES responses (see simulation results in *Figure 6*, upper panel), which was not observed in these brain regions. This model also predicts a stronger quadratic effect of confidence in participants for which the variance ratio between the signal and noise distributions is particularly high. However, the variance ratio was not significantly correlated with the quadratic effect of confidence in any of these regions, as would be expected if they were representing LLR or a similar quantity (see *Appendix 6—figure 1*).

For the next two models, confidence was assumed to be directly proportional to the LLR, with the measured signal representing internal beliefs about hidden model parameters. In the 'Dynamic Criterion' model, we considered whether a quadratic effect of confidence in detection may reflect the active tuning of decision policy in the absence of explicit feedback (*Guggenmos et al., 2016*; *Ko and Lau, 2012*). In the model, beliefs about the underlying distributions are updated on a trial-to-trial basis, and in turn affect the placement of decision criterion (for a formal description of the model, see Appendix section 10). The Dynamic Criterion model predicts that the magnitude of shift in decision criterion will display a positive quadratic relation to confidence (LLR) in detection but not discrimination (see simulation results in *Figure 6*, middle panel). This is because the problem is asymmetric in detection, and decision policy should depend on beliefs about both sensory precision

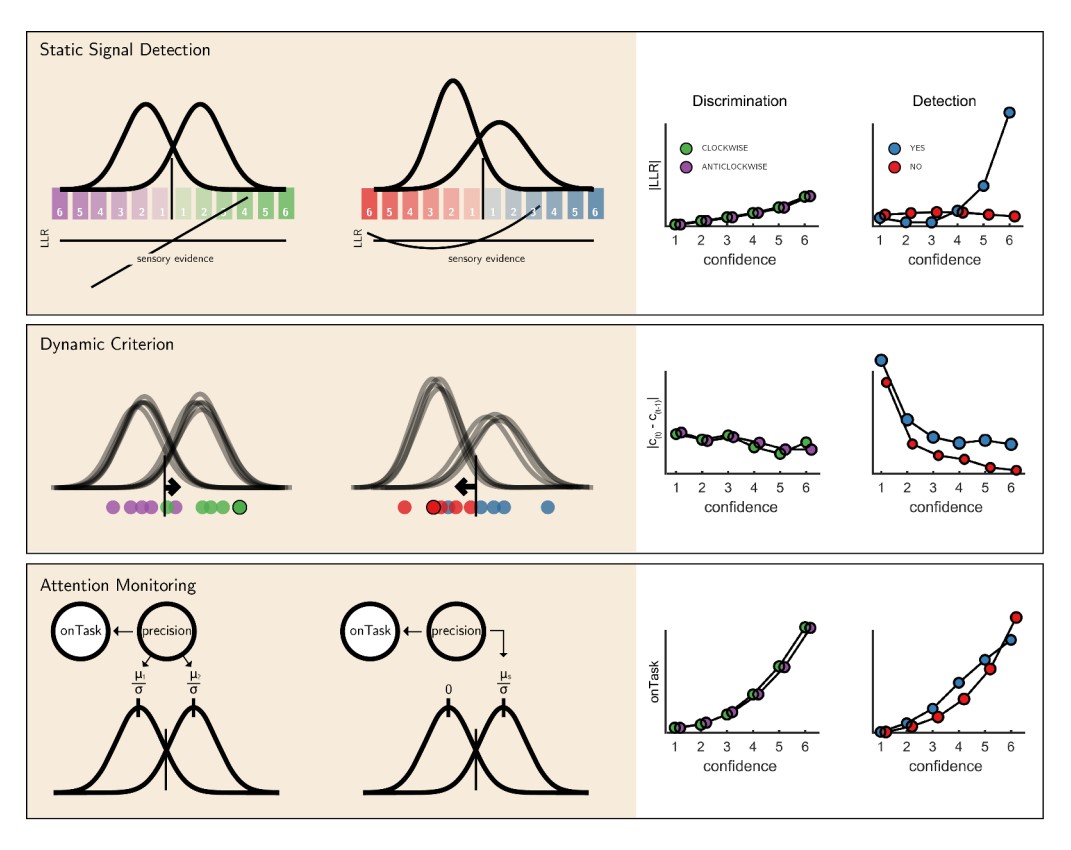

**Figure 6.** The three models (left) and their prediction for confidence effects (right). Top panel: In Signal Detection Theory, perceptual decisions and confidence ratings are generated by comparing the sensory evidence to a fixed set of criteria. In detection the 'signal' distribution is assumed to have higher variance. Plotting the absolute value of the log likelihood ratio as a function of decision and confidence results in a linear curve for discrimination, and a pronounced quadratic effect for YES responses in detection, an effect that is specific to unequal-variance SDT. Middle panel: In a Dynamic Criterion model beliefs about the mean and variance of the perceptual distributions are updated as a function of incoming samples (plotted as circles) and the decision criterion is shifted accordingly. Plotting the absolute change in criterion placement as a function of decision and confidence results in a quadratic effect of confidence for detection responses only. Bottom: In the Attention Monitoring model, beliefs about overall attentiveness ('onTask' node) probabilistically reflect sensory precision. Plotting beliefs about overall attentiveness as a function of decision and confidence results in an overall quadratic effect of confidence, and an interaction between YES and NO responses in detection. For a detailed specification of all three models see appendix sections 9, 10 and 11.

(or the relative variance of the noise and signal distribution) and expected signal strength (mean of the signal distribution), which is not the case for a symmetric discrimination problem.

Notably, the pattern of criterion shifts in the Dynamic Criterion model resembled the task-specific effect of confidence in the FPC, STS and pre-SMA. As a post-hoc test of a role for these regions in criterion adjustment, we examined sequential pairs of trials of the same stimulus category (for example, a signal present trial that was followed by a signal present trial), and contrasted 'repeat' trials with 'switch' trials (for example, [YES, YES] vs. [YES, NO]). The Dynamic Criterion model predicts stronger activation in switch compared to stay trials in both detection and discrimination. The FPl showed a weak effect in this direction ($t = 2.03$, $p=0.05$, $d = 0.34$), whereas FPm, pre-SMA, right BA10 and STS did not (all p-values>0.15).

Finally, we considered a higher-order 'Attention Monitoring' model in which beliefs about one's current attentional state (precision or inverse variance in SDT) are taken into account when making perceptual decisions and confidence ratings on detection trials. This model formalizes the notion that after not detecting a target the participant may ask *'Given my current attentional state, would I have missed the target?'*. The Attention Monitoring model thus makes different predictions for confidence in detection 'target absent' (NO) responses, where the participant is assumed to reflect on the detection-likelihood of hypothetical targets, compared to 'target present' (YES) responses, similar to

the activation profile observed in the rTPJ. However, this model also predicts a pronounced quadratic confidence profile for all four responses, which we do not see in our data.

## Discussion

Previous studies of the neural basis of human perceptual decision-making have tended to focus on discrimination judgments, such as sorting stimuli into category A or B. The general computational architecture supporting discrimination judgments can be naturally extended to support detection (for instance, within signal detection theory). However, computational considerations and behavioral findings suggest that forming confidence in detection judgments may rest on qualitatively distinct cognitive and neural processes in comparison to generating confidence in discrimination judgments.

To test for such differences, here we acquired functional MRI data from 35 participants who reported their subjective confidence in judgments about stimulus type (discrimination), and target presence or absence (detection). These judgments were given on separate trials that were well-matched for stimulus characteristics, response requirements and task difficulty. Across both tasks, we found the expected linear effects of confidence in our pre-specified regions of interest in the prefrontal and parietal cortex. Specifically, in the precuneus, vmPFC, pMFC and ventral striatum, the effect of confidence was invariant to task and response. In contrast, having adjusted our planned design matrix to be sensitive to non-monotonic effects of confidence, we observed a quadratic effect of confidence in detection judgments in the frontopolar cortex (medial and lateral surfaces of BA10), that was absent for discrimination judgments. Similar quadratic activation profiles were observed for both YES and NO responses. Whole-brain analysis revealed a similar effect of task on the quadratic effect of confidence in the right STS and the pre-SMA. Since task performance was matched across the two tasks and since we did not observe overall differences in activation between detection and discrimination (see *Appendix 4—figure 1*), these differences in confidence profiles are unlikely to originate from experimental confounds such as task difficulty, but instead indicate a unique neurocognitive contribution to metacognition of detection judgments. In what follows we will unpack what this contribution might be.

The three regions that showed an interaction of the quadratic expansion of confidence with task in our whole-brain analysis (right frontopolar cortex, right STS, and pre-SMA), as well as two anatomical subcomponents of our frontopolar ROI (FPl and FPm), all shared a very similar activation profile. In detection, the quadratic effect of confidence was positive, but was almost entirely absent for the discrimination task. Follow-up analysis confirmed that this difference was not driven by motor aspects of the confidence rating procedure, such as the number of increase or decrease confidence steps taken to reach the desired confidence level, which was similar for the two tasks (see *Appendix 1—figure 1*). Ours is not the first report of a quadratic relation between activation in prefrontal cortical structures and different subjective ratings. For example, in a study by *Christensen et al. (2006)*, participants were presented with masked stimuli and gave subjective visibility ratings on a three-point scale. The right frontopolar cortex showed decreased activation for 'clear perception' and 'no perception' categories relative to a middle 'vague perception' category. Similarly, *De Martino et al. (2017)* reported a quadratic effect of product desirability in the pMFC. However, for both of the above cases, a quadratic effect can reflect a monotonic relationship with an implicit representation of subjective confidence (*Lebreton et al., 2015*). For example, participants may be more confident in the 'clear perception' and 'no perception' responses compared to the 'vague perception' option, or more confident about liking or not liking a product, compared to when using the middle parts of the liking scale. This explanation cannot account for the observed quadratic trend in our case, where in addition to strong activation levels for the highest confidence ratings in target presence and absence, we also find strong activation levels for the lowest levels of confidence.

We are unable to determine whether this effect originates from one homogeneous population of neurons that shows a quadratic effect of detection confidence, or from two overlapping populations that show nonlinear positive and negative effects of detection confidence – summing to an overall quadratic effect at the voxel level (similar to positive and negative confidence-selective neurons in the human posterior parietal cortex; *Rutishauser et al., 2018*) Addressing this question would require higher spatial resolution, for example using single-cell recordings in patients. Furthermore, because confidence judgments were always preceded by perceptual decisions in our design, we cannot determine whether the observed effects reflect an implicit representation of uncertainty,

computed in parallel with the perceptual decision itself, or a higher-order representation that emerges at the explicit confidence rating phase. Future studies which use model-based estimates of covert decision confidence (*Bang and Fleming, 2018*) or EEG-informed fMRI to resolve early and late processing stages (*Gherman and Philiastides, 2018*) may answer this question.

We considered three alternative computational models that were able to account for asymmetries between detection and discrimination activation profiles. An unequal variance signal detection theory model provided a simple account of the asymmetry between detection and discrimination, but could not account for the similar quadratic profiles observed for YES and NO responses. A more direct test of the proposal that a detection-specific quadratic effect of confidence originates from the unequal-variance properties of stimulus distributions in detection would be to test for similar effects in a discrimination task in which one category of stimuli is of higher variance (e.g., *Denison et al., 2018*). In contrast, the Dynamic Criterion model provided good qualitative accounts for distinct regional activation profiles, and the Attention Monitoring account predicted an interaction between confidence in judgments about presence and absence. However, the Attention Monitoring model also predicted a quadratic effect in discrimination, which we did not see.

Notably, both of these models share the need to learn (in the Dynamic Criterion model) or estimate (in the Attention Monitoring model) the current level of precision (inverse variance) in detection. Such online precision estimation evinces a profound asymmetry between detection and discrimination tasks: in discrimination tasks, one simply has to evaluate the relative evidence for different causes of sensory samples, under some prior belief about sensory precision; namely, the precision of the likelihood that any particular cause (e.g., CLOCKWISE or ANTICLOCKWISE orientation) would generate sensory samples. In contrast, detection presents a difficult (ill-posed, dual estimation) problem. When assessing the evidence for the absence of a target, there could be no sensory evidence because the target is not there or because precision is low (or both). This puts pressure on the estimation of precision to resolve conditional dependencies between posterior beliefs about target presence and the precision with which it can be detected. In short, two things have to be estimated; the posterior expectation about the target and posterior beliefs about precision (*Clark, 2013*; *Feldman and Friston, 2010*; *Haarsma et al., 2018*; *Palmer et al., 2019*; *Parr et al., 2018*).

In line with a role in monitoring of attention or precision, right TPJ showed a negative effect of confidence that was stronger for 'target absent' responses compared to 'target present' responses in detection. This cluster was closest to the posterior subdivision of the right TPJ (TPJp-R; *Igelström et al., 2015*), which is most strongly associated with reasoning about others' beliefs (*Igelström et al., 2016*). In addition to its role in Theory of Mind (*Saxe and Wexler, 2005*; *Lee and McCarthy, 2016*), previous work has highlighted the importance of the rTPJ in controlling attention (*Marois et al., 2004*; *Geng and Vossel, 2013*; *Lee and McCarthy, 2016*; *Dugué et al., 2018*) and filtering distractors in visual search (*Shulman et al., 2007*). Furthermore, damage to the rTPJ can result in visual hemineglect: a condition in which stimuli in the left visual hemifield fail to reach awareness (*Corbetta et al., 2005*). Together, these observations have led to a proposal (the 'Attention Schema Theory') that the rTPJ is maintaining a simplified representation of one's own and others' attentional states, and that this function makes this region essential for maintaining conscious awareness (*Graziano and Webb, 2015*).

The current Attention Monitoring model fits well with the Attention Schema Theory. A representation of one's current attentional state is a useful source of information for determining confidence in detection judgments, because stimuli are more likely to be missed when participants are not paying careful attention. This will be specifically useful for judgments about stimulus absence: if a target was not observed, the participant may reason something along the lines of *'given my current state of attention, I was not very likely to miss a target, therefore I can be very confident that a target was not presented'*. In support of this idea, the typically poor metacognitive evaluations of decisions about stimulus absence are partially recovered when task difficulty is controlled by manipulating attention rather than stimulus visibility (*Kanai et al., 2010*; *Kellij et al., 2018*), suggesting that subjects may harness information about their attentional state to inform their confidence judgments. Interestingly, the frontopolar cortex, which showed a detection-specific quadratic effect of confidence in our experiment, has also been implicated in attentional control via the gating of internal and external modes of attention (*Burgess et al., 2007*) and in discriminating between imagined and externally perceived memory items (*Simons et al., 2006*; *Turner et al., 2008*). Together, the

engagement of this set of regions in detection confidence hints at a potential role for self-monitoring of attention in metacognition of detection.

To conclude, we find a quadratic effect of confidence in detection judgments in several brain regions, including the frontopolar cortex and rTPJ. In the frontopolar cortex, this quadratic effect was not seen for discrimination judgments. In the rTPJ, we also found a linear effect of confidence that was more negative for judgments about stimulus absence compared to judgments about stimulus presence. We consider three computational accounts of our results, two of which implicate the learning and estimation of signal-to-noise statistics as promising accounts of the observed detection-specific activation profiles. However, while each of these accounts could explain some of our findings, none of the models could provide a complete account of the data. Further work is needed to decide between these alternatives, or to suggest new ones.

## Materials and methods

All design and analysis details were pre-registered before data acquisition and time-locked using pre-RNG randomization (*Mazor et al., 2019*). The time-locked protocol folder is available at https://github.com/matanmazor/detectionVsDiscrimination_fMRI (*Mazor, 2020*; copy archived at https://github.com/elifesciences-publications/detectionVsDiscrimination_fMRI). The entire set of pre-registered analyses results is available at https://osf.io/98mv4/.

### Participants

46 participants took part in the study (ages 18–36, mean = 24 ± 4; 29 females). 35 participants met our pre-specified inclusion criteria (ages 18–36, mean = 24 ± 4; 20 females). After applying our run-wise exclusion criteria to the data of the remaining 35 participants, our dataset consisted of 5 usable experimental runs from 15 participants, 4 usable experimental runs from 14 participants, 3 usable experimental runs from 5 participants, and 2 usable experimental runs from one participant. We pre-specified a sample-size of 35, balancing statistical power and resource considerations.

### Design and procedure

After a temporally jittered rest period of 500–4000 milliseconds, each trial started with a fixation cross (500 milliseconds), followed by a presentation of a target for 33 milliseconds. In discrimination trials, the target was a circle of diameter 3° containing randomly generated white noise, merged with a sinusoidal grating (2 cycles per degree; oriented 45° or −45°). In half of the detection trials, targets did not contain a sinusoidal grating and consisted of random noise only. After stimulus offset, participants used their right-hand index and middle fingers to make a perceptual decision about the orientation of the grating (discrimination blocks), or about the presence or absence of a grating (detection blocks). The response mapping was counterbalanced between blocks, such that an index finger press was used to indicate a CLOCKWISE tilt on half of the trials, and an ANTICLOCKWISE tilt on the other half. Similarly, in half of the detection trials the index finger was mapped to a YES ('target present') response, and on the other half to a NO ('target absent') response.

Immediately after making a decision, participants rated their confidence on a 6-point scale by using two keys to increase and decrease their reported confidence level with their left-hand thumb. Confidence levels were indicated by the size and color of a circle presented at the center of the screen. The initial size and color of the circle was determined randomly at the beginning of the confidence rating phase, to decorrelate the number of button presses and the final confidence rating. The mapping between color and size to confidence was counterbalanced between participants: for half of the participants high confidence was mapped to small, red circles, and for the other half high confidence was mapped to large, blue circles. This counterbalancing was employed to isolate confidence-related activations from activations that originate from the perceptual properties of the confidence scale or from differences in the motor requirement to press the upper and lower buttons. The perceptual decision and the confidence rating phases were restricted to 1500 and 2500 milliseconds, respectively. No feedback was delivered to subjects about their performance.

Participants were acquainted with the task in a preceding behavioral session. During this session, task difficulty was adjusted independently for detection and for discrimination, targeting around 70% accuracy on both tasks. We achieved this by adaptively controlling the stimulus signal-to-noise ratio (SNR) once in every 10 trials: increasing the SNR when accuracy fell below 60%, and decreasing

it when accuracy exceeded 80%. Performance on the detection and discrimination task was further calibrated to the scanner environment at the beginning of the scanning session, during the acquisition of anatomical (MP-RAGE and fieldmap) images. After completing the calibration phase, participants underwent five ten-minute functional scanner runs, each comprising one detection and one discrimination block of 40 trials each, presented in random order.

To avoid stimulus-driven fluctuations in confidence, grating SNR was fixed within each experimental block. Nevertheless, following experimental blocks with markedly bad ($\leq$ 52.5%) or good ($\geq$ 85%) accuracy, grating SNR was adjusted for the next block of the same task (SNR level was divided or multiplied by a factor of 0.9 for bad and good performance, respectively). Finally, grating SNR was adjusted for both tasks following runs in which the difference in performance between the two tasks exceeded 16.25% (SNR level was multiplied by the square root of 0.9 for the easier task and divided by the square root of 0.9 for the more difficult task).

To incentivize participants to do their best at the task and rate their confidence accurately, we offered a bonus payment according to the following payment schedule: bonus = £$\frac{\overrightarrow{accuracy} \cdot \overrightarrow{confidence}}{200}$

Where $\overrightarrow{accuracy}$ is a vector of 1 and $-1$ for correct and incorrect responses, and $\overrightarrow{confidence}$ is a vector of integers in the range of 1 to 6, representing confidence reports for all trials. We explained the payment structure to participants in the preceding behavioral session. Specifically, we advised participants that to maximize their bonus they should do their best at the main task, rate the confidence higher when they believe they are correct, and rate their confidence lower when they believe they might be wrong.

## Scanning parameters

Scanning took place at the Wellcome Centre for Human Neuroimaging, London, using a 3 Tesla Siemens Prisma MRI scanner with a 64-channel head coil. We acquired structural images using an MPRAGE sequence ($1 \times 1 \times 1$ mm voxels, 176 slices, in plane FoV = $256 \times 256$ mm$^2$), followed by a double-echo FLASH (gradient echo) sequence with TE1 = 10 ms and TE2 = 12.46 ms (64 slices, slice thickness = 2 mm, gap = 1 mm, in plane FoV = $192 \times 192$ mm$^2$, resolution = $3 \times 3$ mm$^2$) that was later used for field inhomogeneity correction. Functional scans were acquired using a 2D EPI sequence, optimized for regions near the orbito-frontal cortex ($3 \times 3 \times 3$ mm voxels, TR = 3.36 s, TE = 30 ms, 48 slices tilted by $-30$ degrees with respect to the T > C axis, matrix size = $64 \times 72$, Z-shim = $-1.4$).

## Analysis

The preregistered objectives of this study were to:

1. Replicate findings of a generic (task-invariant) confidence signal in the activity of medial prefrontal cortex (**De Martino et al., 2013**; **Morales et al., 2018**).
2. Test for an interaction between the parametric effect of confidence level and task (detection/discrimination) in the BOLD response in prefrontal cortex ROIs.
3. Within detection trials, test for an interaction between the parametric effect of confidence level and response (YES/NO) in the BOLD response, specifically in the prefrontal cortex and in frontopolar regions that have previously been associated with counterfactual reasoning (**Boorman et al., 2009**; **Donoso et al., 2014**).
4. Test for relationships between fluctuations in metacognitive adequacy (a trial-by-trial measure of metacognitive sensitivity; **Wokke et al., 2017**), and the BOLD signal separately for detection and for discrimination, and for YES and NO responses within detection.
5. Replicate previous findings of between-subject correlations between lateral prefrontal cortex (lPFC) function and metacognitive efficiency (meta-d'/d'; **Fleming and Lau, 2014**) in discrimination (**Yokoyama et al., 2010**).
6. Identify between-subject functional correlates of metacognitive efficiency in detection. Specifically, ask if metacognitive efficiency in detection is predicted by activity in distinct networks compared to metacognitive efficiency in discrimination.

## Exclusion criteria

Subjects were excluded from all analyses for any of the following pre-specified reasons: missing more than 20% of the trials, performing one of the tasks with accuracy below 60%, exceeding the 4

mm affine motion cutoff criterion in more than 2 experimental runs, and showing a consistent response bias (i.e. using the same response in more than 75% of the trials) in at least one task. Individual scan runs were excluded from all analyses if the participant exceeded the affine motion cutoff, if more than 20% of trials were missed, if mean accuracy was below 60% or if the response bias for one of the tasks exceeded 80%.

In addition, we applied a confidence-related exclusion criterion: participants were excluded if they used the same confidence level in more than 80% of all trials globally or for a particular response, and individual scan runs were excluded if the same confidence level was used in more than 95% of the trials, either globally or for particular response types. Our preregistration document specified that the confidence exclusion criterion will be used to exclude participants from confidence-related analyses only, but we subsequently revised this plan in order to use identical design matrices for all participants.

## Behavioral analysis
### Response conditional type-II ROC curves
Response conditional type-II ROC (Receiver Operating Characteristic) curves were extracted for the two discrimination and two detection responses. This was done by plotting the cumulative distribution of confidence levels in correct responses against the cumulative distribution of confidence levels in incorrect responses. As a measure of response-specific metacognitive sensitivity, we extracted the area under these curves (*AUROC2*). The expected AUROC2 for no metacognitive insight (i.e., the confidence distributions are identical for correct and incorrect responses) is 0.5. Perfect metacognitive insight (i.e., confidence in all correct responses is higher than confidence in all incorrect responses) will result in an AUROC2 of 1.

## Imaging analysis
### fMRI data preprocessing
Data preprocessing followed the procedure described in *Morales et al. (2018)*: 'Imaging analysis was performed using SPM12 (Statistical Parametric Mapping; www.fil.ion.ucl.ac.uk/spm). The first five volumes of each run were discarded to allow for T1 stabilization. Functional images were realigned and unwarped using local field maps (*Andersson et al., 2001*) and then slice-time corrected (*Sladky et al., 2011*). Each participant's structural image was segmented into gray matter, white matter, CSF, bone, soft tissue, and air/background images using a nonlinear deformation field to map it onto template tissue probability maps (*Ashburner and Friston, 2005*). This mapping was applied to both structural and functional images to create normalized images in Montreal Neurological Institute (MNI) space. Normalized images were spatially smoothed using a Gaussian kernel (6 mm FWHM). We set a within-run 4 mm affine motion cutoff criterion'.

Preprocessing and construction of first- and second-level models used standardized pipelines and scripts available at https://github.com/metacoglab/MetaLabCore/.

## Regions of interest
In addition to an exploratory whole-brain analysis (corrected for multiple comparisons at the cluster level), our analysis focused on the following a priori regions of interest, largely following the ROIs used by *Fleming et al. (2018)*:

1. Frontopolar cortex (FPC, defined anatomically). We used a connectivity-based parcellation (*Neubert et al., 2014*) to define a general FPC region of interest as the total area spanned by areas FPl, FPm and BA46. The right hemisphere mask was mirrored to create a bilateral mask.
2. Ventromedial prefrontal cortex (vmPFC). The vmPFC ROI was defined as a 8 mm sphere around MNI coordinates [0,46,–7], obtained from a meta-analysis of subjective-value related activations (*Bartra et al., 2013*) and aligned to the cortical midline.
3. Bilateral ventral striatum. The ventral striatum ROIs was specified anatomically from the Oxford-Imanova Striatal Structural Atlas included with FSL (http://fsl.fmrib.ox.ac.uk).
4. Posterior medial frontal cortex (pMFC). The pMFC ROI was defined as a 8 mm sphere around MNI coordinates [0, 17, 46], obtained from a functional MRI study on decision confidence and aligned to the cortical midline (*Fleming et al., 2012*).

5. Precuneus. The precuneus ROI was defined as a 8 mm sphere around MNI coordinates [0,– 57,18], based on Voxel- Based Morphometry studies of metacognitive efficiency (*Fleming et al., 2010*; *McCurdy et al., 2013*) and aligned to the cortical midline.

For the general FPC ROI, small-volume correction was applied to individual voxels within the ROI for all univariate contrasts. For the multivariate analysis, we used a searchlight approach to scan for spatial patterns within the ROI, followed by a correction for multiple comparisons. For all other ROIs, a GLM was fitted to the mean time course of voxels within the region, and multivariate analysis was performed on all voxels within the ROI. While our pre-registered analysis defined the frontopolar cortex as a single region, we subsequently decided to separately analyze its 3 separate anatomical subregions identified by *Neubert et al. (2014)* (FPl, FPm and BA46). The decision to separate the FPC ROI to its subcomponents was made after data collection and these anatomical subregions should not be taken as a priori ROIs.

## Univariate analysis

Univariate analysis was based on a design matrix in which different trial types are modeled by different regressors (main design matrix, below). Additionally, to examine the global effect of confidence across trial types, a simpler design matrix was fitted to the data as a first step (global confidence design matrix, below). Experimental runs for each subject were temporally concatenated before estimating the GLM coefficients. This was done in order to maximize sensitivity to response- and task-specific modulations of confidence, given the limited and varying number of trials within each experimental run.

### Main design matrix (DM-1)

The main design matrix for the univariate GLM analysis consisted of 16 regressors of interest. There was a regressor for each of the eight combinations of task x condition x response: For example, a regressor for detection trials where a signal was present and the subject reported seeing a signal with a YES response (present and present, P_P). The relevant trials were modeled by a boxcar regressor with nonzero entries at the interval starting at the offset of the stimulus and ending immediately after the confidence rating phase, convolved with the canonical hemodynamic response function (HRF). The duration of this interval was 4300 milliseconds, and not 4000 milliseconds as mistakenly indicated in the preregistration document. Each of these primary regressors was accompanied by a linear parametric modulation of the confidence reported for each trial. Together, the design matrix included 16 regressors of interest (see *Table 1*).

Trials in which the participant did not respond within the 1500 millisecond time frame were modeled by a separate regressor. The design matrix also include a run-wise constant term regressor, an instruction-screen regressor for the beginning of each block, motion regressors (the 6 motion parameters and their first derivatives as extracted by SPM in the head motion correction preprocessing phase) and regressors for physiological measures. Button presses were modeled as stick functions, convolved with the canonical HRF, in three regressors: two regressors for the right and left right-hand buttons, and one regressor for both up and down left-hand presses. We decided to have one regressor for both types of left-hand presses due to the strong positive correlation of the final confidence rating with the number of 'increase confidence' button presses, and the strong negative correlation with the number of 'decrease confidence' button presses.

### Global confidence design matrix (GC-DM)

The global confidence design matrix consisted of 4 regressors of interest. The first two primary regressors were 'correct trials' (trials in which the participant was correct, across tasks and responses) and 'incorrect trials' (trials in which the participant was incorrect, across tasks and responses). Single events were modeled by a boxcar regressor with nonzero entries at the 4300 millisecond interval starting at the offset of the stimulus and ending immediately after the confidence rating phase, convolved with the canonical hemodynamic response function (HRF). Additionally, the design matrix included a confidence parametric modulator for each of the first two regressors. The construction of the regressors and the additional nuisance regressors was handled similarly to the main design.

**Table 1.** List of regressors in the main design matrix (DM-1).

| | | Task | Stimulus | Response |
|---|---|---|---|---|
| 1 | CW_CW | Discrimination | Clockwise | Clockwise |
| 2 | CW_CW_conf | | | |
| 3 | CW_ACW | Discrimination | Clockwise | Anticlockwise |
| 4 | CW_ACW_conf | | | |
| 5 | ACW_CW | Discrimination | Anticlockwise | Clockwise |
| 6 | ACW_CW_conf | | | |
| 7 | ACW_ACW | Discrimination | Anticlockwise | Anticlockwise |
| 8 | ACW_ACW_conf | | | |
| 9 | P_P | Detection | Present | Present |
| 10 | P_P_conf | | | |
| 11 | P_A | Detection | Present | Absent |
| 12 | P_A_conf | | | |
| 13 | A_P | Detection | Absent | Present |
| 14 | A_P_conf | | | |
| 15 | A_A | Detection | Absent | Absent |
| 16 | A_A_conf | | | |

## Quadratic-Confidence design matrix (post-hoc analysis; QC-DM)

The quadratic-confidence design matrix for the univariate GLM analysis consisted of 12 regressors of interest. There was a regressor for each of the four responses: YES, NO, CLOCKWISE and ANTICLOCKWISE. Similar to the main design matrix, the relevant trials were modeled by a boxcar regressor with non-zero entries at the 4300 millisecond interval starting at the offset of the stimulus and ending immediately after the confidence rating phase, convolved with the canonical hemodynamic response function (HRF). Each of these primary regressors was accompanied by two parametric modulators, representing the linear and quadratic effects of confidence. Together, the design matrix included 12 regressors (4 responses + 4 linear confidence regressors + 4 quadratic confidence regressors). The QC-DM included the same set of nuisance regressors as the main design matrix.

## Categorical-Confidence design matrices (post-hoc analysis; CC-DM)

In order to better understand the nature of the linear interaction between confidence in YES and NO responses, we specified a pair of design matrices—one for each task—in which confidence level was modeled as a categorical variable. Instead of the 8 primary regressors in the main design matrix, this design matrix consisted of only one regressor of interest for all trials, modeled by a boxcar with non-zero entries at the 4300 millisecond interval starting at the offset of the stimulus and ending immediately after the confidence rating phase, convolved with the canonical hemodynamic response function (HRF). This regressor was in turn modulated by a series of 12 dummy (0/1) parametric modulators - one for every response (YES and NO for detection and CLOCKWISE and ANTICLOCKWISE for discrimination) and confidence rating (1–6 for both tasks). Using two design matrices instead of one allowed us to set discrimination trials to be the baseline category for detection, and detection trials as the baseline for discrimination. These design matrices included the same set of nuisance regressors as the main design matrix.

For each participant, we used the beta-estimates from the categorical-confidence design matrices as the input to four response-specific multiple linear regression models, with linear confidence and quadratic confidence as predictors, in addition to an intercept term. The subject-specific coefficients were then subjected to ordinary least squares group-level inference, to compare linear and quadratic effects of confidence between responses. The rationale for choosing this two-step approach was its indifference to the confidence distributions for the four responses, that may bias the estimation of the quadratic and linear terms.

## Multivariate analysis

Multi-voxel pattern analysis (*Norman et al., 2006*) was used to test for consistent spatial patterns in the fMRI data. We used The Decoding Toolbox (*Hebart et al., 2015*) and followed the procedures described by *Morales et al. (2018)*. In order to identify brain regions that are implicated in inference about presence and absence, we trained and tested a linear classifier on detection decisions. We classified hits and correct rejections, instead of hits and misses as originally planned, due to an insufficient number of detection misses in some experimental blocks. We then compared the resulting classification accuracy with the cross-classification accuracy of training on detection responses and testing on discrimination confidence and vice versa. The purpose of this comparison was to isolate neural correlates of inference about stimulus absence or presence that should be specific to detection from more general neural correlates of stimulus visibility, that are also expected to affect confidence in discrimination judgements (see *Appendix 8—figure 1*).

The other prespecified multivariate tests were designed to find universal and response-specific spatially multivariate representations of confidence. After conducting this analysis we came to realize that our experimental design was not appropriate for estimating the degree to which the representation of confidence is 'response-general'. In our experimental design, confidence is confounded with visual feedback during the confidence-rating phase, such that 'response-general' representations of confidence could appear if the spatial pattern of activation was sensitive to the visual feedback in the confidence rating. For completeness, we include the results of this analysis in the osf project page, but do not interpret them further.

## Statistical inference

T-test and anova Bayes factors use a Jeffrey-Zellner-Siow Prior for the null distribution, with a unit prior scale (*Rouder et al., 2009*; *Rouder et al., 2012*). Whole-brain fMRI significance was corrected for family-wise error rate at the cluster level (p<0.05), with a cluster defining threshold of p<0.001.

## Acknowledgements

We thank Maayan Keshev, Dan Bang, Madeleine Scott, Peter Zeidman, Nadège Corbin, Tim Tierney, Emma Holmes, Max Rollwage, Roni Maimon, Rani Moran, Noam Mazor and the FIL imaging team for their help in different stages of this project. The Wellcome Centre for Human Neuroimaging is supported by core funding from the Wellcome Trust (203147/Z/16/Z). SMF is supported by a Sir Henry Dale Fellowship jointly funded by the Wellcome Trust and the Royal Society (206648/Z/17/Z).

## Additional information

### Funding

| Funder | Grant reference number | Author |
| --- | --- | --- |
| Royal Society | Sir Henry Dale Fellowship 206648/Z/17/Z | Stephen M Fleming |
| Wellcome Trust | Sir Henry Dale Fellowship 206648/Z/17/Z | Stephen M Fleming |

The funders had no role in study design, data collection and interpretation, or the decision to submit the work for publication.

### Author contributions

Matan Mazor, Conceptualization, Data curation, Formal analysis, Validation, Investigation, Visualization, Methodology, Writing - original draft, Project administration, Writing - review and editing; Karl J Friston, Software, Supervision, Writing - review and editing; Stephen M Fleming, Conceptualization, Resources, Supervision, Funding acquisition, Validation, Investigation, Methodology, Project administration, Writing - review and editing

## Author ORCIDs
Matan Mazor (iD) https://orcid.org/0000-0002-3601-0644
Karl J Friston (iD) http://orcid.org/0000-0001-7984-8909
Stephen M Fleming (iD) https://orcid.org/0000-0003-0233-4891

## Ethics
Human subjects: Participants gave their informed consent to take part in the experiment. The experiment was approved by the UCL ethics committee (approval numbers 8231/001 and 1260/003).

## Decision letter and Author response
Decision letter https://doi.org/10.7554/eLife.53900.sa1
Author response https://doi.org/10.7554/eLife.53900.sa2

## Additional files
### Supplementary files
• Transparent reporting form

### Data availability
Group statistical parametric maps are available on NeuroVault:https://identifiers.org/neurovault.collection:6065. Single subject behavioural data and functional data from our regions of interest is available on: https://github.com/matanmazor/detectionVsDiscrimination_fMRI (copy archived at https://github.com/elifesciences-publications/detectionVsDiscrimination_fMRI). Single subject raw anonymized data is available upon request.

The following datasets were generated:

| Author(s) | Year | Dataset title | Dataset URL | Database and Identifier |
|---|---|---|---|---|
| Mazor M, Friston KJ, Fleming SM | 2020 | Confidence in Detection and Discrimination | https://identifiers.org/neurovault.collection:6065 | NeuroVault, 6065 |
| Mazor M, Fleming S, Friston K | 2018 | Detection and Discrimination Imaging | https://osf.io/98mv4/ | Open Science Framework, 10.17605/OSF.IO/98MV4 |

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

## Appendix 1

### Confidence button presses

**Appendix 1—figure 1.** Average number of button presses for each confidence level, as a function of task. More button presses were needed on average to reach the extreme confidence ratings, hence the quadratic shape. No difference between the two tasks was observed in the mean number of button presses for any of the confidence levels. Error bars represent the standard error of the mean.

# Appendix 2

## zROC curves

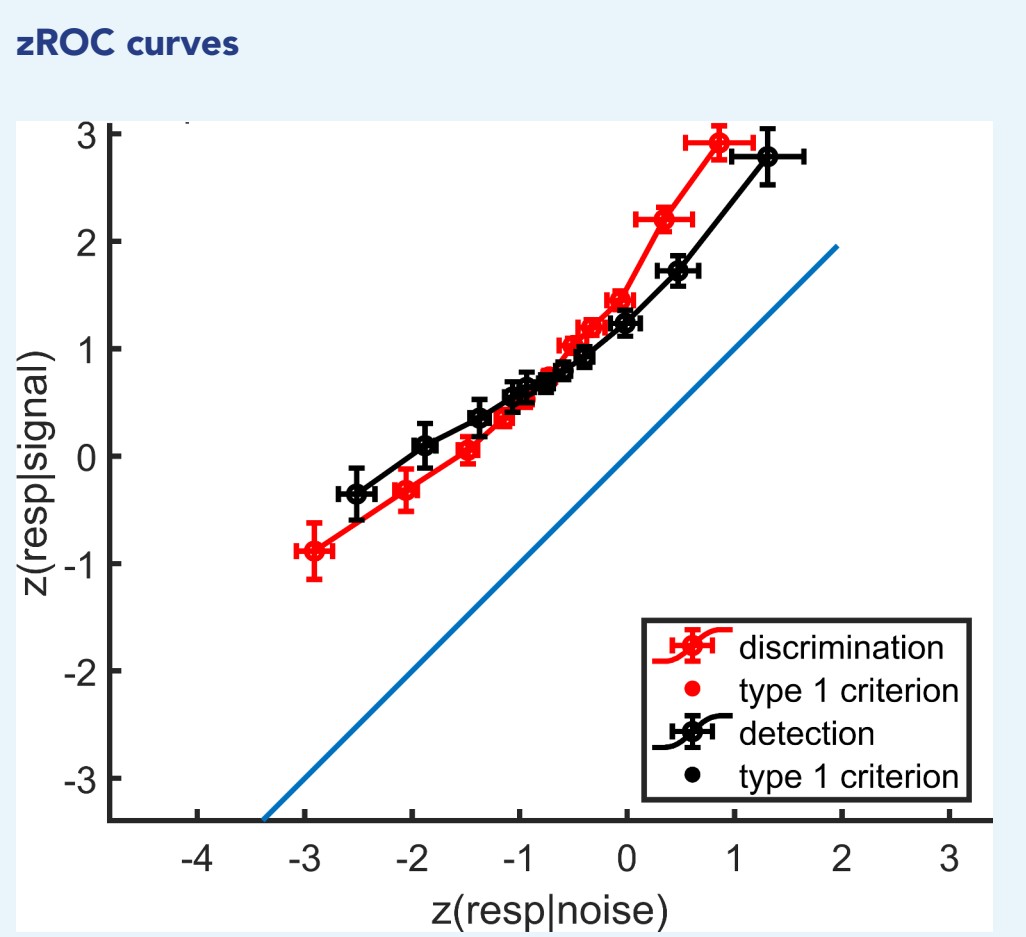

**Appendix 2—figure 1.** mean zROC curves for the discrimination and detection tasks. As expected in a uv-SDT setting, the discrimination curve is approximately linear with a slope of 1, and the detection curve is approximately linear with a shallower slope. Error bars represent the standard error of the mean.

## Appendix 3

### Global confidence design matrix

From our pre-specified ROIs, only the vmPFC and BA46 ROIs showed a significant linear effect of confidence in correct responses, in the opposite direction to what we expected based on previous studies. This is likely to be due to the differences in confidence profiles between the detection and discrimination tasks:

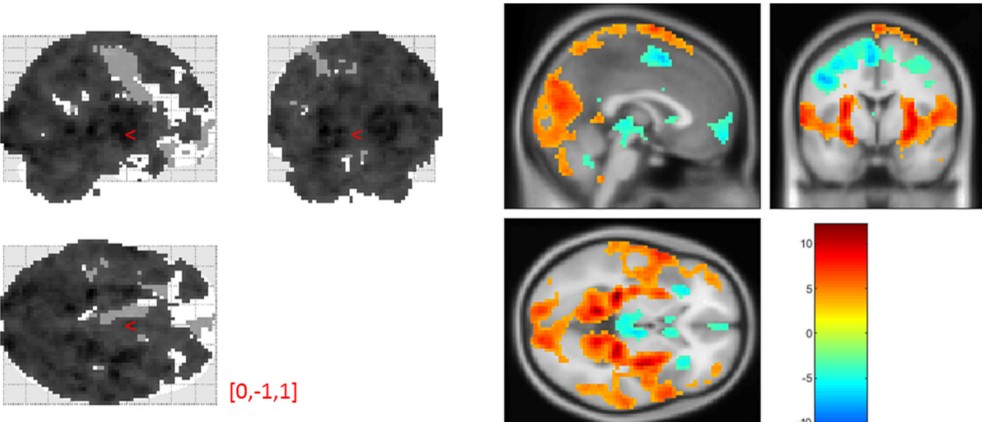

**Appendix 3—figure 1.** Effect of confidence in correct responses, from the global-confidence design matrix. Uncorrected, thresholded at p<0.001. Left: glass brain visualization of the whole brain contrast. Right: yellow-red represent a positive correlation with subjective confidence ratings, and green-blue represent a negative correlation.

| | Average beta | T value | P value | Standard deviation |
|---|---|---|---|---|
| vmPFC | -0.35 | -3.06 | $4 \times 10^{-3}$ | 0.67 |
| pMFC | -0.31 | -2.48 | 0.02 | 0.74 |
| precuneus | 0.25 | 2.30 | 0.03 | 0.64 |
| ventral striatum | -0.056 | -1.51 | 0.14 | 0.22 |
| FPl | 0.16 | 1.52 | 0.14 | 0.64 |
| FPm | -0.12 | -1.46 | 0.16 | 0.48 |
| BA 46 | 0.37 | 3.77 | $6 \times 10^{-4}$ | 0.57 |

## Appendix 4

### Main effect of task

Main effect of task (detection-discrimination). Main design matrix

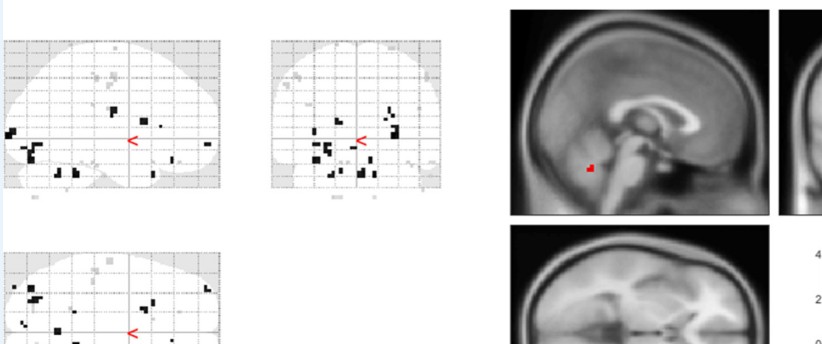

[0,-1,1]

**Appendix 4—figure 1.** main effect of task, from the main design matrix. Uncorrected, thresholded at p<0.001. Left: glass brain visualization of the whole brain contrast. Right: yellow-red represent stronger activations in detection, and green-blue in discrimination. None of our ROIs showed a main effect of task (detection vs. discrimination).

|  | Average beta | T value | P value | Standard deviation |
|---|---|---|---|---|
| vmPFC | -0.01 | -0.05 | 0.96 | 1.64 |
| pMFC | 0.15 | 0.60 | 0.55 | 1.45 |
| precuneus | -0.04 | -0.16 | 0.87 | 1.65 |
| ventral striatum | 0.09 | 0.77 | 0.45 | 0.72 |
| FPl | 0.28 | 1.08 | 0.29 | 1.55 |
| FPm | $5 \times 10^{-3}$ | 0.02 | 0.98 | 1.22 |
| BA 46 | 0.38 | 1.19 | 0.24 | 1.89 |

## Appendix 5

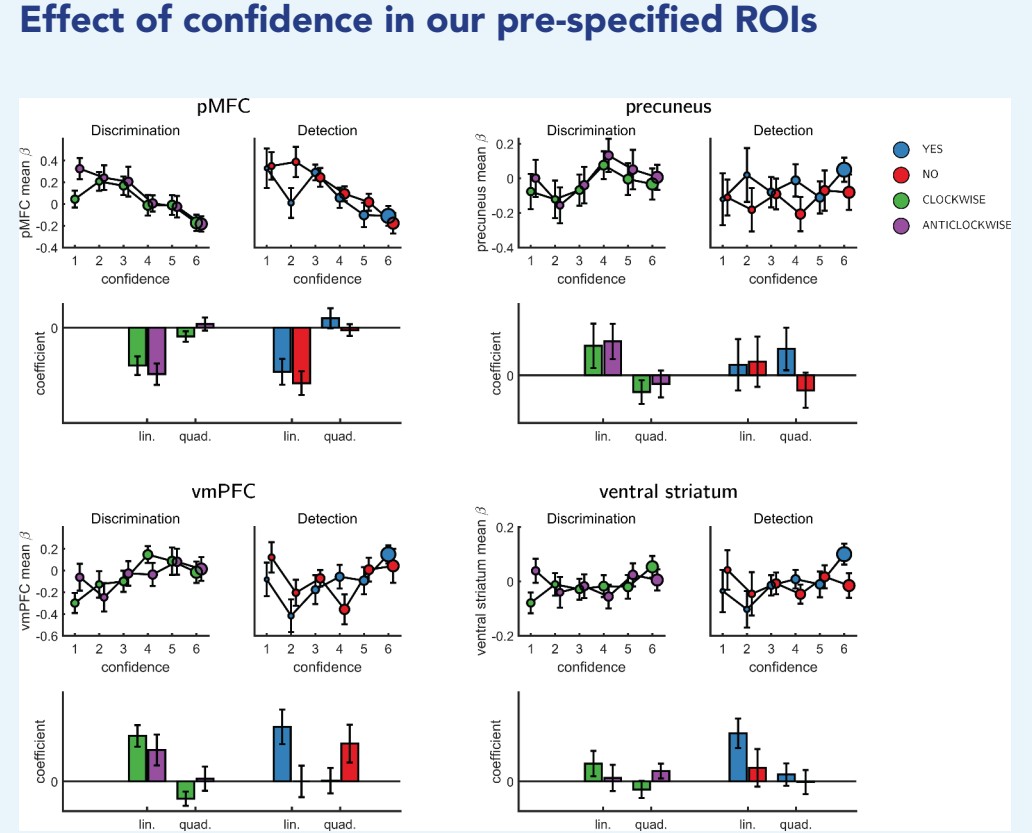

**Appendix 5—figure 1.** Effect of confidence in all 4 ROIs, as a function of task and response, as extracted from the categorical design matrix. No significant interaction between the linear or quadratic effects and task or response was observed in any of the ROIs.

## Appendix 6

### SDT variance ratio correlation with the quadratic confidence effect

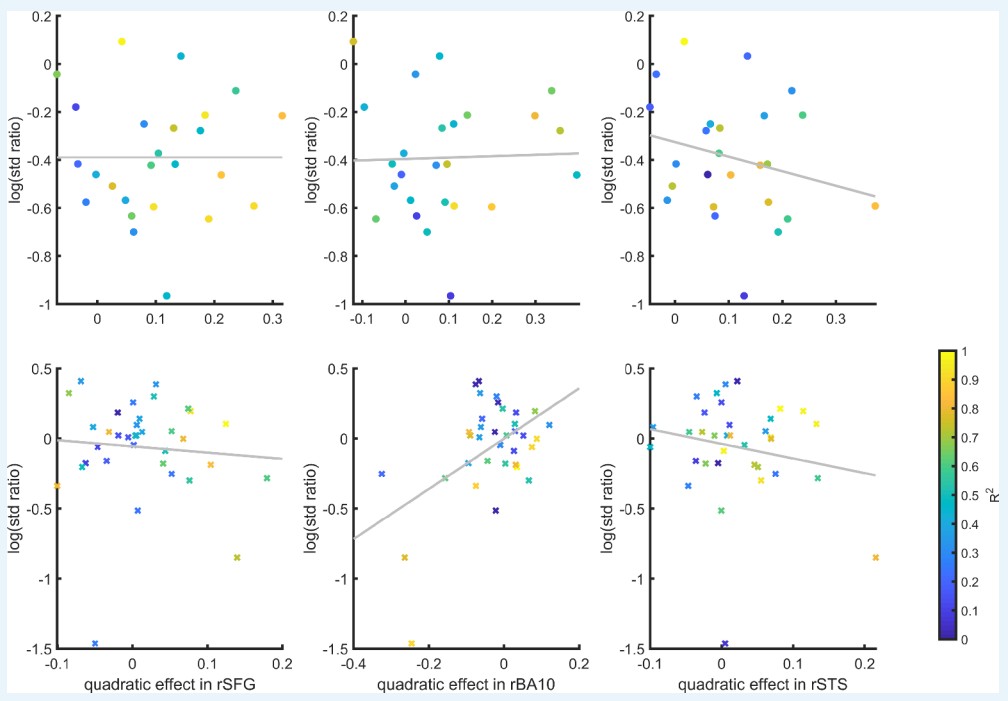

**Appendix 6—figure 1.** Inter-subject correlation between the quadratic effect in the right hemisphere clusters and the ratio between the detection (top panel) and discrimination (lower panel) distribution variances, as estimated from the zROC curve slopes in the two tasks. Marker color indicates the goodness of fit of the second-order polynomial model to the BOLD data. All Spearman correlation coefficients are <0.25.

## Appendix 7

### Correlation of metacognitive efficiency with linear and quadratic confidence effects

**Appendix 7—figure 1.** Inter-subject correlation between the linear (upper panel) and quadratic (lower panel) effects in the right hemisphere clusters and the metacognitive efficiency scores (measured as M ratio = meta-d′/d′, *Maniscalco and Lau, 2012*).

## Appendix 8

### Confidence-decision cross classification

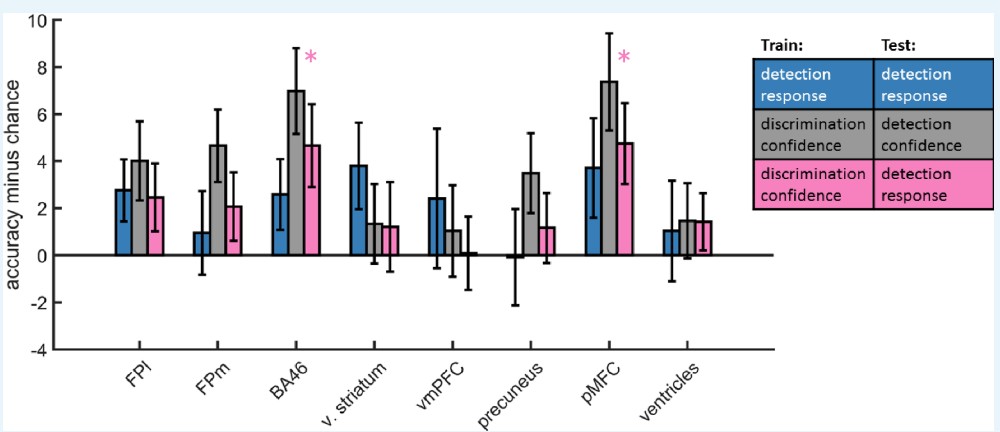

**Appendix 8—figure 1.** Accuracy minus chance for classification of response in detection (YES vs. NO; blue), and from a cross-classification between tasks: confidence in detection and confidence in discrimination (gray), and confidence in discrimination and decision in detection (pink). In order to dissociate between brain regions that encode stimulus visibility and brain regions that encode decision confidence, we performed a multivariate cross-classification analysis. We trained a linear classifier on detection decisions (YES and NO), and tested it on discrimination confidence (high and low), and vice versa. Shared information content between detection responses and confidence in discrimination is expected in brain regions that encode stimulus visibility, rather than accuracy estimation. In detection, YES responses are associated with higher stimulus visibility compared to NO responses (regardless of decision confidence), and in discrimination high confidence trials are associated with higher visibility than low confidence trials (regardless of subjective confidence).

Presented cross classification scores are the mean of cross classification accuracies in both directions. Detection-response and discrimination-confidence cross-classification was significantly above chance in in the pMFC (t(29)=2.76, p<0.05, corrected for family-wise error across the four ROIs), and in the BA46 anatomical subregion of the frontopolar ROI (t(29)=2.64, p<0.05, corrected).

## Appendix 9

# Static signal detection theory

## Discrimination

### Generative model

According to SDT, a decision variable $x$ is sampled from one of two distributions on each experimental trial.

$$\mu_t = \begin{cases} 0.5, & \text{if CW.} \\ -0.5, & \text{if ACW.} \end{cases} \tag{1}$$

$$x_t \sim \mathcal{N}(\mu_t, 1) \tag{2}$$

### Inference

$x$ is compared against a criterion to generate a decision about which of the two distributions was most likely, given the sample. For a discrimination task with equally likely symmetric distributions around 0, the optimal placement for a criterion is at 0.

$$decision_t = \begin{cases} \text{CW}, & \text{if } x_t > 0. \\ \text{ACW}, & \text{else.} \end{cases} \tag{3}$$

In standard discrimination tasks, a common assumption is that the two distributions are Gaussian with equal variance. This assumption has a convenient computational consequence: the log-likelihood ratio (LLR), a quantity that reflects the degree to which the sample is more likely under one distribution or another, is linear with respect to $x$. Confidence is then assumed to be proportional to the distance of $x_t$ from the decision criterion.

In what follows $\phi(x, \mu, \sigma)$ is the likelihood of observing x when sampling from a normal distribution with mean $\mu$ and standard deviation $\sigma$.

$$LLR = log(\phi(x_t, 0.5, 1)) - log(\phi(x_t, -0.5, 1)) \tag{4}$$

$$confidence_t \propto |x_t| \tag{5}$$

## Detection

### Generative model

A common assumption is that in detection the signal distribution is wider than the noise distribution (unequal-variance SDT; **Wickens, 2002**, section 3.4).

$$\mu_t = \begin{cases} 1.3, & \text{if P.} \\ 0, & \text{if A.} \end{cases} \tag{6}$$

$$\sigma_t = \begin{cases} 2, & \text{if P.} \\ 1, & \text{if A.} \end{cases} \tag{7}$$

$$x_t \sim \mathcal{N}(\mu_t, \sigma_t) \tag{8}$$

### Inference

Here $med(x)$ represents the median sensory sample $x$. This criterion was chosen to ensure that detection responses are balanced.

$$decision = \begin{cases} \text{P}, & \text{if } x_t > med(x). \\ \text{A}, & \text{else.} \end{cases} \qquad (9)$$

Importantly, in uv-SDT, LLR is quadratic in x.

$$LLR = log(\phi(x, 1.3, 2)) - log(\phi(x, 0, 1)) \qquad (10)$$

$$confidence \propto |x_t - med(x)| \qquad (11)$$

**Appendix 10**

## Dynamic criterion

In SDT, task performance depends on the degree of overlap between the underlying distributions (d') and on the positioning of the decision criterion (c). Participants may optimize criterion placement based on their changing beliefs about the underlying distributions (*Lau, 2007*; *Ko and Lau, 2012*). To model this dynamic process of criterion setting we simulated a model where beliefs about the underlying distributions are the Maximum Likelihood Estimates of the mean and standard deviation, based on the last 5 samples that were (correctly or not) categorized.

### Discrimination

#### Generative Model

As in the Static Signal Detection model.

#### Inference

Means and standard deviations of the two distributions are estimated based on the last 5 samples in each category. To model prior beliefs about these parameters, each participant starts the task with 5 imaginary samples from the veridical distributions. Means and standard deviations are then extracted from these imaginary samples. In what follows, $\vec{cw}$ and $\vec{acw}$ are vectors with entries corresponding to the last 5 samples that were (correcly or not) labelled as CLOCKWISE and ANTICLOCKWISE, respectively. $\bar{x}_{cw}$ and $\bar{x}_{acw}$ correspond to the sample means of these vectors. $\sigma_{cw}$ and $\sigma_{acw}$ correspond to their standard deviations.

$$LLR = log(\phi(x, \bar{x}_{cw}, \sigma_{cw})) - log(\phi(x, \bar{x}_{acw}, \sigma_{acw})) \tag{12}$$

Decisions and confidence are extracted from the *LLR* as in the Static Signal Detection model.

### Detection

#### Generative Model

As in the Static Signal Detection model.

#### Inference

As in discrimination. In what follows, $\vec{a}$ and $\vec{p}$ are vectors with entries corresponding to the last 5 samples that were (correcly or not) labelled as 'signal absent' and 'signal present', respectively. $\bar{x}_a$ and $\bar{x}_p$ correspond to the sample means of these vectors. $\sigma_a$ and $\sigma_p$ correspond to their standard deviations.

$$LLR = log(\phi(x, \bar{x}_p, \sigma_p)) - log(\phi(x, \bar{x}_a, \sigma_a)) \tag{13}$$

In detection, *LLR* = 0 at two points (see *Figure 6*). The decision criterion $c_t$ is chosen to coincide with the rightmost point, which is positioned between the Signal and Noise distribution means.

$$decision = \begin{cases} P, & \text{if } x_t > c_t. \\ A, & \text{else.} \end{cases} \tag{14}$$

$$confidence \propto |LLR| \tag{15}$$

## Appendix 11

### Attention monitoring

Similar to the Dynamic Criterion model, in the Attention Monitoring model participants adjust a decision criterion based on changing beliefs about the underlying distributions. However, unlike the Dynamic Criterion model, here beliefs change not as a function of recent perceptual samples, but as a function of access to an internal variable that represents the expected sensory precision (attention).

#### Discrimination

##### Generative model

In our schematic formulation of this model, participants have a true attentional state, which for simplicity we treat as either being on (1) or off (0). When attending, participatns enjoy higher sensitivity than when they are not attending.

$$p(attended_t) = 0.5 \tag{16}$$

The attentional state determines the means of sensory distributions.

$$\mu_t = \begin{cases} 0.5, & \text{if CW and } \neg attended_t. \\ -0.5, & \text{if ACW and } \neg attended_t. \\ 2, & \text{if CW and } attended_t. \\ -2, & \text{if ACW and } attended_t. \end{cases} \tag{17}$$

$$x_t \sim \mathcal{N}(\mu_t, 1) \tag{18}$$

However, they do not have direct access to their attentional state, but only to a noisy approximation of the probability that they were attending.

$$onTask_t \sim \begin{cases} Beta(2,1), & \text{if } attended_t. \\ Beta(1,2), & \text{if } \neg attended_t. \end{cases} \tag{19}$$

##### Inference

Participants are then assumed to use their knowledge about the *onTask* variable when making a decision and confidence estimate.

$$\begin{aligned} p(x_t|\text{CW}) &= p(attended_t|onTask_t)\phi(x_t, 2, 1) + p(\neg attended_t|onTask_t)\phi(x_t, 0.5, 1) \\ &= onTask_t\phi(x_t, 2, 1) + (1 - onTask_t)\phi(x_t, 0.5, 1) \end{aligned} \tag{20}$$

$$\begin{aligned} p(x_t|\text{ACW}) &= p(attended_T|onTask_t)\phi(x_t, -2, 1) + p(\neg attended_t|onTask_t)\phi(x_t, -0.5, 1) \\ &= onTask_t\phi(x_t, -2, 1) + (1 - onTask_t)\phi(x_t, -0.5, 1) \end{aligned} \tag{21}$$

$$LLR = log(p(x_t|\text{CW})) - log(p(x_t|\text{ACW}) \tag{22}$$

$$decision_t = \begin{cases} \text{CW}, & \text{if } LLR > 0. \\ \text{ACW}, & \text{else.} \end{cases} \tag{23}$$

$$confidence_t \propto |LLR| \tag{24}$$

## Detection

### Generative model

In detection, attentional states only affect the signal distribution, as noise is always centred at 0.

$$\mu_t = \begin{cases} 0, & \text{if A and } \neg attended_t. \\ 0.5, & \text{if P and } \neg attended_t. \\ 0, & \text{if A and } attended_t. \\ 2, & \text{if P and } attended_t. \end{cases} \tag{25}$$

$$x_t \sim \mathcal{N}(\mu_t, 1) \tag{26}$$

### Inference

$$\begin{aligned} p(x_t|\text{P}) &= p(attended_t|onTask_t)\phi(x_t, 2, 1) + p(\neg attended_t|onTask_t)\phi(x_t, 0.5, 1) \\ &= onTask_t\phi(x_t, 2, 1) + (1 - onTask_t)\phi(x_t, 0.5, 1) \end{aligned} \tag{27}$$

The likelihood of observing $x_t$ if no stimulus was presented is independent of the attention state.

$$\begin{aligned} p(x_t|\text{A}) &= p(attended_t|onTask_t)\phi(x_t, 0, 1)) + p(\neg attended_t|onTask_t)\phi(x_t, 0, 1) \\ &= \phi(x_t, 0, 1) \end{aligned} \tag{28}$$

$$LLR = log(p(x_t|\text{p})) - log(p(x_t|\text{a})) \tag{29}$$

$$decision_t = \begin{cases} \text{P}, & \text{if } LLR > 0. \\ \text{A}, & \text{else.} \end{cases} \tag{30}$$

Nevertheless, confidence in judgments about stimulus absence is dependent on beliefs about the attentional state. This is mediated by the effect of attention on the likelihood of observing $x_t$ if a stimulus were present. This is the counterfactual part.

$$confidence_t \propto |LLR| \tag{31}$$

