## [Decision Letter]

**Acceptance summary:**

This study explores whether neural correlates of metacognitive judgments differ between detection and discrimination tasks. Results show that neural correlates of confidence in fronto-polar cortex differ by task, such that activity correlates non-linearly with confidence in detection tasks. In addition, the authors show that right temporoparietal junction is uniquely recruited for absence judgments.

**Decision letter after peer review:**

Thank you for submitting your article "Distinct neural contributions to metacognition for detecting (but not discriminating) visual stimuli" for consideration by *eLife*. Your article has been reviewed by three peer reviewers, one of whom is a member of our Board of Reviewing Editors, and the evaluation has been overseen by Joshua Gold as the Senior Editor. The following individual involved in review of your submission has agreed to reveal their identity: Michael Graziano (Reviewer #3).

The reviewers have discussed the reviews with one another and the Reviewing Editor has drafted this decision to help you prepare a revised submission.

Summary:

This study explores the neural correlates of metacognitive judgments of detection compared to discrimination. The authors find that confidence in detection judgments differs from that of discrimination in frontopolar cortex, and that right TPJ may be involved in confidence for absence judgments. Most importantly, they show that activity in detection judgements has a quadratic relationship with confidence ratings. All reviewers agreed that the paper is interesting and informative. However, there were also questions regarding the Discussion and whether alternative explanations have been adequately explored.

Essential revisions:

1) More needs to be said in the Discussion about what the quadratic confidence effects really mean. Do these signals originate from neurons that show a quadratic relationship between firing and confidence, or do they come from two overlapping populations that show positive and negative linear relationships with confidence, respectively? In the former case, what do neurons that respond to only to very high and very low confidence really signal. To these neurons encode confidence as such or a function of confidence? For instance, such a function could be the confidence of the confidence rating, which, as for any subjective report, should be higher at the extremes of the scale. However, in this case, it is the confidence about the confidence rating not the confidence about the perceptual judgement. The question is whether you would still observe quadratic confidence effects in detection task if there was no confidence rating? The authors bring up the related concept of meta-confidence as it can be derived from the Bayesian inference model. But it is not made clear enough that here confidence is on precision not on the detection judgement itself. These issues of meta-confidence need to be discussed more expansively and also independently of the Bayesian model.

2) Related to this, in several areas the quadratic effect is stronger in the detection than in the discrimination task. So this says that these regions care about whether you're really confident, in which case you should probably learn something about your environment based on feedback regarding the outcome of your choice (e.g. Guggenmos et al., 2016), or you're really not confident, in which case you should not update your model of the world at all. So what are these areas coding for? How much you care to update your world model based on the outcome of a choice? What does the quadratic relationship buy the organism?

3) Still related to the interpretation of the quadratic effect, it would be helpful to expand on the models and mechanisms that can give rise to nonlinear confidence effects. For instance, the suggestion that equal vs. unequal variance of the sample in the SDT model can explain linear vs. nonlinear effects is very interesting but this needs to be unpacked and explained in much more detail. The same is true (though to a lesser degree) for the Bayesian model. It may be helpful to include a figure illustrating how different models can explain nonlinear confidence effects. In general, these theoretical/conceptual considerations are an essential part of the current study and it would be important for the authors to expand on this.

4) The difference in response-conditional metacognitive sensitivity between yes and no responses, as shown also by Meuwese et al., and the lack of difference between YES-conditional metacognitive sensitivity and discrimination-based metacognitive sensitivity, suggests that the difference in metacognitive sensitivity may result from difference in variance of the underlying internal response distributions. If YES- and discrimination-dependent metacognitive sensitivity is similar, and the only one that is different is no-dependent metacognitive sensitivity, doesn't that imply simply that the signal distribution has larger variance than the noise across both tasks? Some simulations could be done to evaluate this possibility. This would be in line with a report from Kellij et al. (2018) on Psyarxiv (doi: 10.31234/osf.io/xky38), suggesting that asymmetric variance is wholly responsible for differences in t2AUC.

5) The exploratory analysis sought regions in which the quadratic effect of confidence was stronger in the detection than the discrimination task. What about the other way around? Were there ROIs where the quadratic effect of confidence was stronger for discrimination than detection?

6) Was the difference between quadratic effects in detection versus discrimination related to the difference in the (negative) linear effects, in any of these ROIs? In other words, is this just a regression to the mean problem, where you're having trouble finding areas that show either linear or quadratic effects of confidence? It seems that the explanation offered in the Discussion may be of utility here, such that if one specifies the SDT system in LLR space there is a quadratic relationship between the internal estimate x and the LLR is quadratic for unequal variance systems (Discussion, eighth paragraph). Although the measure of variance inequality across individuals was not correlated with the quadratic effect of confidence in the reported ROIs, I wonder if this might be mediated by the SNR of the BOLD signal in each individual, which could maybe be informed by the relationship between the linear differences between detection and discrimination and their quadratic relationship difference. In other words, if the linear relationship is weak in a given person, that could also imply a weaker quadratic relationship due to irrelevant factors such as SNR of the BOLD signal. Unless I am missing a point, this could destroy cross-subject relationships between zROC-based estimates of variance inequality and quadratic magnitude. At the least, the authors could test whether the variance imbalanced revealed by the slope of the zROC is related to any of these measures.

---

## [Author Response]

Essential revisions:1) More needs to be said in the Discussion about what the quadratic confidence effects really mean. Do these signals originate from neurons that show a quadratic relationship between firing and confidence, or do they come from two overlapping populations that show positive and negative linear relationships with confidence, respectively? In the former case, what do neurons that respond to only to very high and very low confidence really signal. To these neurons encode confidence as such or a function of confidence? For instance, such a function could be the confidence of the confidence rating, which, as for any subjective report, should be higher at the extremes of the scale. However, in this case, it is the confidence about the confidence rating not the confidence about the perceptual judgement. The question is whether you would still observe quadratic confidence effects in detection task if there was no confidence rating? The authors bring up the related concept of meta-confidence as it can be derived from the Bayesian inference model. But it is not made clear enough that here confidence is on precision not on the detection judgement itself. These issues of meta-confidence need to be discussed more expansively and also independently of the Bayesian model.

We thank the reviewers for prompting further reflection on the interpretation of the confidence response profiles here. The observed activation profile in the FPC, STS and pre-SMA could indeed originate from one homogeneous population of neurons that shows a quadratic effect of confidence, or from two overlapping populations that show a nonlinear positive and negative effects of confidence – summing to an overall quadratic effect at the voxel level. It is difficult to evaluate these alternatives with the current design, as they do not make different predictions at the level of the BOLD signal. We are also unable to tell whether these activations would continue to be observed in the absence of an explicit confidence rating, given that a rating was always required in the current design. However, the quadratic response profile cannot be explained as a result of simply using a confidence scale: if this were the case, we would have observed the same pattern in the discrimination trials which also required an explicit subjective judgment. We have now added the following section to the Discussion to make clear where further work is needed to evaluate these alternatives:

“We are unable to determine whether this effect originates from one homogeneous population of neurons that shows a quadratic effect of detection confidence, or from two overlapping populations that show nonlinear positive and negative effects of detection confidence – summing to an overall quadratic effect at the voxel level (similar to positive and negative confidence-selective neurons in the human posterior parietal cortex; Rutishauser et al., 2015). […] Future studies which use model-based estimates of covert decision confidence (Bang and Fleming, 2018) or EEG-informed fMRI to resolve early and late processing stages (Gherman and Philiastides, 2018) may answer this question.”

In response to point (3) below, we now also unpack in greater detail the possible computational mechanisms that may have given rise to a quadratic profile.

2) Related to this, in several areas the quadratic effect is stronger in the detection than in the discrimination task. So this says that these regions care about whether you're really confident, in which case you should probably learn something about your environment based on feedback regarding the outcome of your choice (e.g. Guggenmos et al., 2016), or you're really not confident, in which case you should not update your model of the world at all. So what are these areas coding for? How much you care to update your world model based on the outcome of a choice? What does the quadratic relationship buy the organism?

We agree with the reviewers that this asymmetry in the quadratic profile with respect to confidence is the key result of our paper. The notion that this is related to updating of a model of the task, or learning, is interesting to pursue – and forms the basis of new simulations that we present in response to point (3) below (please see below).

3) Still related to the interpretation of the quadratic effect, it would be helpful to expand on the models and mechanisms that can give rise to nonlinear confidence effects. For instance, the suggestion that equal vs. unequal variance of the sample in the SDT model can explain linear vs. nonlinear effects is very interesting but this needs to be unpacked and explained in much more detail. The same is true (though to a lesser degree) for the Bayesian model. It may be helpful to include a figure illustrating how different models can explain nonlinear confidence effects. In general, these theoretical/conceptual considerations are an essential part of the current study and it would be important for the authors to expand on this.

We agree and are glad of the opportunity to expand on potential models of the quadratic effect. We now simulate three different models that predict detection-specific non-linear effects of confidence: a Signal Detection model, a Dynamic Criterion model, and an Attention Monitoring model. We include simulations from the three models in the appendix and as part of our GitHub repository here: https://github.com/matanmazor/detectionVsDiscrimination_fMRI/tree/master/simulation

We describe the three models and their predictions in a new section of the Results section (subsection “Computational models”).and include Figure 6 for clarity.

4) The difference in response-conditional metacognitive sensitivity between yes and no responses, as shown also by Meuwese et al., and the lack of difference between YES-conditional metacognitive sensitivity and discrimination-based metacognitive sensitivity, suggests that the difference in metacognitive sensitivity may result from difference in variance of the underlying internal response distributions. If YES- and discrimination-dependent metacognitive sensitivity is similar, and the only one that is different is no-dependent metacognitive sensitivity, doesn't that imply simply that the signal distribution has larger variance than the noise across both tasks? Some simulations could be done to evaluate this possibility. This would be in line with a report from Kellij et al. (2018) on Psyarxiv (doi: 10.31234/osf.io/xky38), suggesting that asymmetric variance is wholly responsible for differences in t2AUC.

We thank the reviewer for suggesting this analysis. Because we used two independent staircase procedures, stimulus visibility (SNR) in detection ‘signal’ trials was generally higher than in discrimination. For this reason, signal variance in detection cannot be directly extrapolated from signal variance in discrimination in the same way as Kellij et al. However, we carried out an additional analysis of the detection and discrimination zROC curves to evaluate extent of variance asymmetries, which we now include as Appendix 2—figure 1. For both tasks the zROC curves were relatively linear, showing a good fit to the SDT assumptions. The discrimination task zROC had a linear slope of approximately ~1, supporting an equal-variance model, whereas the detection task zROC showed a shallower slope of < 1, consistent with an uv-SDT model. However, uv-SDT does not preclude interpretations of the metacognitive disparity effect at the decision-making or metacognitive levels, especially in light of the sensitivity of this effect to the means by which stimuli are made difficult to perceive (Kanai et al., 2011; Kellij et al., 2018).

5) The exploratory analysis sought regions in which the quadratic effect of confidence was stronger in the detection than the discrimination task. What about the other way around? Were there ROIs where the quadratic effect of confidence was stronger for discrimination than detection?

We did not find any brain region that showed stronger effects of confidence (linear or quadratic) for discrimination over detection. This point is now clarified in the manuscript:

“…voxels, peak voxel: [9,65,-10], Z=4.00). Importantly, no region showed stronger quadratic effects of confidence in discrimination compared to detection.”

6) Was the difference between quadratic effects in detection versus discrimination related to the difference in the (negative) linear effects, in any of these ROIs? In other words, is this just a regression to the mean problem, where you're having trouble finding areas that show either linear or quadratic effects of confidence? It seems that the explanation offered in the Discussion may be of utility here, such that if one specifies the SDT system in LLR space there is a quadratic relationship between the internal estimate x and the LLR is quadratic for unequal variance systems (Discussion, eighth paragraph). Although the measure of variance inequality across individuals was not correlated with the quadratic effect of confidence in the reported ROIs, I wonder if this might be mediated by the SNR of the BOLD signal in each individual, which could maybe be informed by the relationship between the linear differences between detection and discrimination and their quadratic relationship difference. In other words, if the linear relationship is weak in a given person, that could also imply a weaker quadratic relationship due to irrelevant factors such as SNR of the BOLD signal. Unless I am missing a point, this could destroy cross-subject relationships between zROC-based estimates of variance inequality and quadratic magnitude. At the least, the authors could test whether the variance imbalanced revealed by the slope of the zROC is related to any of these measures.

We thank the reviewer for suggesting this useful control analysis. We agree that cross-subject correlation may indeed have been masked by differences in overall noisiness of the signal between participants, especially with this relatively modest sample size for examining between-subject effects. We have now tackled this potential concern in a new analysis. To estimate the noisiness of single subject data, we extracted subject-wise R-squared for the second-order polynomial model predicting BOLD signal from confidence level and response. We replotted Appendix 6—figure 1, this time with a color-code that indicates this goodness of fit for each subject. If the relation between zROC slope and the quadratic coefficient is masked by variability in overall data quality, a correlation should be unmasked when focusing on only the participants with high R-squared scores. We did not see evidence for such an effect in any of the three clusters – instead, the relationship with log(SD ratio) seemed similar for different subgroups of subjects.